# Nephron progenitor commitment is a stochastic process influenced by cell migration

Kynan T Lawlor[1], Luke Zappia[1,2], James Lefevre[3], Joo-Seop Park[4], Nicholas A Hamilton[3], Alicia Oshlack[1,2], Melissa H Little[1,5,6]*, Alexander N Combes[1,6]*

[1]Murdoch Children's Research Institute, Parkville, Australia; [2]School of Biosciences, University of Melbourne, Melbourne, Australia; [3]Division of Genomics of Development and Disease, Institute for Molecular Bioscience, The University of Queensland, Brisbane, Australia; [4]Division of Pediatric Urology and Division of Developmental Biology, Cincinnati Children's Hospital Medical Center, University of Cincinnati College of Medicine, Cincinnati, United States; [5]Department of Paediatrics, The University of Melbourne, Melbourne, Australia; [6]Department of Anatomy and Neuroscience, University of Melbourne, Melbourne, Australia

**Abstract** Progenitor self-renewal and differentiation is often regulated by spatially restricted cues within a tissue microenvironment. Here, we examine how progenitor cell migration impacts regionally induced commitment within the nephrogenic niche in mice. We identify a subset of cells that express *Wnt4*, an early marker of nephron commitment, but migrate back into the progenitor population where they accumulate over time. Single cell RNA-seq and computational modelling of returning cells reveals that nephron progenitors can traverse the transcriptional hierarchy between self-renewal and commitment in either direction. This plasticity may enable robust regulation of nephrogenesis as niches remodel and grow during organogenesis.
DOI: https://doi.org/10.7554/eLife.41156.001

**\*For correspondence:**
Melissa.Little@mcri.edu.au (MHL);
alexander.combes@unimelb.edu.au (ANC)

## Introduction

Mammalian kidney morphogenesis is propagated by reciprocal signalling between progenitor cell populations at the periphery of the developing kidney (*Combes et al., 2015*; *Costantini and Kopan, 2010*; *Kopan et al., 2014*; *Little and McMahon, 2012*). Nephron progenitor cells play a central role in this process, signalling to the tips of the ureteric epithelium to promote branching morphogenesis, and providing a pool of cells from which nephron formation can be induced (*Combes et al., 2015*; *Costantini and Kopan, 2010*; *Kobayashi et al., 2008*). Iterative rounds of ureteric branching and nephron induction continue until the nephron progenitor population differentiates near the end of gestation (*Hartman et al., 2007*; *Rumballe et al., 2011*; *Ryan et al., 2018*). The resulting complement of nephrons underpins renal function in adult life.

Much progress has recently been made in understanding the signals that regulate nephron progenitor fate (*Brown et al., 2013*; *Brown et al., 2015*; *Combes et al., 2015*; *Li et al., 2016*; *Park et al., 2012*; *Park et al., 2007*). It is less clear how individual nephron progenitors interact with these signals in a three dimensional context. Nephron progenitor cells surround the tips of the ureteric epithelium forming a 'cap mesenchyme' domain that is embedded within, but distinct from cortical stroma and vasculature. Together these populations form the nephrogenic niche, with the least committed nephron progenitor cells at the 'top' of the tip and nephron formation occurring at the 'bottom', in the junction between tip and stalk of the ureteric epithelium (*Mugford et al., 2009*).

The spatial segregation of uncommitted and committing nephron progenitor cells is thought to be dictated by differences in signalling activity in the tip versus the stalk of the ureteric epithelium. Tips produce WNT11, BMP7, FGF9 and WNT9B, which act to maintain progenitor self-renewal and survival (*Barak et al., 2012*; *Brown et al., 2013*; *Karner et al., 2011*; *Majumdar et al., 2003*). Canonical Wnt signalling, involving WNT9B from the ureteric epithelium, is critical to induce the epithelialisation of nephron progenitor cells to form nephrons (*Carroll et al., 2005*; *Karner et al., 2011*; *Park et al., 2007*; *Ramalingam et al., 2018*). The prevailing model of nephron progenitor differentiation is that cells close to the tip-stalk junction receive higher levels of WNT9B and initiate commitment by upregulating further genes required for nephron initiation such as *Wnt4* (*Kispert et al., 1998*; *Stark et al., 1994*). Prior to epithelialisation, nephron progenitors coalesce to form a pretubular aggregate (PTA), which is characterised as a cluster of cells in the tip-stalk junction, defined by expression of *Wnt4, Fgf8, Pax8,* and *Lhx1* (*Carroll et al., 2005*; *Georgas et al., 2009*; *Stark et al., 1994*). Detailed studies of cell polarity and lumen formation in the early nephron identify PTAs as groups of cells within the tip-stalk junction that do not have a lumen or defined apical-basal polarity (*Yang et al., 2013*). Cells within the PTA transition to a primitive renal vesicle (RV), defined as having one or two apical foci containing polarity proteins such as aPKC and PAR3. These foci connect to form a single continuous lumen in a mature renal vesicle, which now represents an epithelium (*Yang et al., 2013*). Patterning and specification of nephron segment identity starts during the formation of these early nephron structures to eventually result in a mature segmented nephron (*Georgas et al., 2009*; *Lindström et al., 2018a*).

Clonal lineage tracing of nephron progenitor cells suggests that one sibling can remain in the progenitor domain while another contributes to a nephron (*Kobayashi et al., 2008*). How one sibling cell commits while the other self-renews is not understood. At a population level, there is support for division of nephron progenitor cells into spatially restricted subdomains that reflect a linear progression in commitment from a self-renewing (*Six2 +Cited1+*), to a primed (*Six2+*), and committed state (*Wnt4 +Lef1+*) (*Brown et al., 2013*; *Brown et al., 2015*; *Carroll et al., 2005*; *Mugford et al., 2009*; *Park et al., 2007*). As such, we might expect that individual progenitors undergo a sequential restriction in potential as they gradually move towards the site of nephron formation and ultimately contribute to an epithelializing PTA. Counter to this model, our own live imaging of 852 individual nephron progenitor cells over 18 hr revealed substantial cell migration within the cap mesenchyme (*Combes et al., 2016*). Nephron progenitor cells moved randomly within and between niches, with this movement influenced by attraction, repulsion, and adhesion to the ureteric tip. Far from a population gradually transitioning through static subdomains, some cells made large movements from the 'self-renewing' to the 'committing' region and vice versa, potentially crossing previously proposed domain boundaries (*Combes et al., 2016*). These stochastic migration events likely vary the exposure of individual nephron progenitor cells to signalling environments within the niche, raising the question of how cell fate is controlled in such a dynamic environment.

In this study, we re-examine nephron progenitor commitment using lineage tracing, live imaging, and computational modelling to reconcile the spatially restricted process of commitment with a motile nephron progenitor population. Using an inducible *Wnt4*-Cre, we find that onset of *Wnt4* expression in the early stages of nephron formation does not always trigger differentiation. A subset of cells that express *Wnt4* at the tip-stalk junction migrate out of this region to re-enter the nephron progenitor domain. While these cells have expressed *Wnt4*, they cease to do so after returning to the progenitor domain and may remain within this region long-term, returning to a transcriptional state equivalent to uncommitted nephron progenitor cells. In contrast, the first commitment events in the developing kidney generate a population of cells that can remain in the tip-stalk region and contribute to multiple nephrons. Integrating the observed nephron progenitor migration and plasticity, we define a model of commitment where stochastic cell movement influences exposure to differentiation cues and determines whether cells remain captured within the nephron induction zone or 'escape' back to the cap mesenchyme. This insight provides a mechanism to explain how a swarm of motile progenitors can both maintain ureteric branching and give rise to committing cells in a manner that is spatially defined and sensitive to changing niche cues.

## Results

### *Wnt4* lineage tracing labels a population of nephron progenitor cells across time

Nephron progenitors are assumed differentiate in a linear fashion from an uncommitted, to a primed then committed state. To investigate this process in more detail, we assessed the differentiation status of individual nephron progenitor cells using *Wnt4* expression as a marker of commitment. We used *Wnt4*$^{GCE}$ mice that encode GFP-fused to CreER$^{T2}$ under control of the endogenous *Wnt4* promoter (*Kobayashi et al., 2008*). To determine whether expression of the GFP-CreER$^{T2}$ element replicated the expected expression pattern of *Wnt4* in the early nephron, we cross-referenced *Wnt4*-GFP to the localisation of polarity protein aPKC, which has been used to characterise the stages of nephron epithelialisation (*Yang et al., 2013*). *Wnt4* expression was first observed in cells at the tip-stalk junction that represent PTA structures prior to epithelialisation. Expression was maintained into the primitive and maturing RV (*Figure 1a–c*). GFP signal was not observed within nephron progenitor cells on top of the tip. *Wnt4*-GFP was evident at lower levels in some stromal cells, particularly in the medulla, consistent with previous reports (*Georgas et al., 2008*; *Itäranta et al., 2006*) (*Figure 1—figure supplement 1*). *Wnt4*$^{GCE}$ mice were crossed to a Cre inducible Rosa26-LSL-tdTomato reporter (*Madisen et al., 2010*). In these embryos, GFP marks cells that currently express *Wnt4*, while *Wnt4*-expressing cells and their descendants are permanently marked by tamoxifen dependant CreER$^{T2}$ activation of tdTomato (*Figure 1d*). Tamoxifen administration results in less than 24 hours (hr) of CreER$^{T2}$ activity (*Kobayashi et al., 2008*). Embryos were labelled at embryonic day (E) 12.5 and kidneys collected at E13.5, E14.5 and E15.5 and stained for tdTomato, GFP and the nephron progenitor marker SIX2 (*Figure 1d–f*). Whole or partial-organ confocal images were captured with volumes typically 200–400 µm in depth, enabling analysis of at least 30–50% of the entire progenitor population per organ (*Figure 1f*).

At E13.5, 24 hr after tamoxifen treatment, tdTomato-labelled *Wnt4* lineage cells were restricted to regions of *Wnt4*-GFP expression in the PTA and RV structures, as anticipated for committing cells (*Figure 1g–k*). Thus tdTomato labelling driven by *Wnt4*$^{GCE}$ preferences the sites of highest *Wnt4* expression (PTA and RV), and rarely labelled stromal cells in these experiments. Cells in the PTA expressed low levels of SIX2 and *Wnt4*-GFP, but *Wnt4*-GFP was not observed in the upper region of the niche (*Figure 1g*). At E14.5 and E15.5, 48 and 72 hr after tamoxifen treatment, tdTomato labelling was observed within developing nephrons as a result of earlier labelling in precursor structures (*Figure 1l–u*). However, we also identified tdTomato-labelled SIX2$^+$ cells within the cap mesenchyme. These appeared as single cells or small groups of cells in some but not all cap domains at E14.5 (*Figure 1l–p*). Such *Wnt4*-lineage progenitors were more apparent at E15.5 with some cap domains containing clusters of tdTomato-labelled cells expressing SIX2 but not *Wnt4*-GFP (*Figure 1q–u*, *Video 1*), suggesting that they were labelled as a result of previous *Wnt4* expression in a PTA rather than induction of differentiation in individual cells away from the site of nephron formation.

We performed lineage tracing with an inducible *Pdgfra* Cre (*Ding et al., 2013*) to test the possibility of a stromal *Wnt4*-lineage contribution to the nephron progenitor population. *Pdgfra* is expressed in all stromal cells in the developing kidney, including those that express *Wnt4* (*DiRocco et al., 2013*). Using this line results in extensive stromal labelling but does not result in labelled cells within the cap mesenchyme (*Figure 1—figure supplement 1*). As such, *Wnt4*-lineage contributions to the nephron progenitor population arise from the established sites of *Wnt4* expression in the early committing nephron, not from the stroma.

We hypothesised that labelled nephron progenitor cells arise from a portion of the committing population that 'escape' commitment and re-enter the cap mesenchyme over time. To test whether these cells were able to self-renew long term, or were prone to differentiation, we labelled at E12.5 and assessed developing kidneys seven days later at E19.5. *Wnt4*-lineage cells were observed in the cap mesenchyme of these samples suggesting that they return to an uninduced progenitor state (*Figure 2a*).

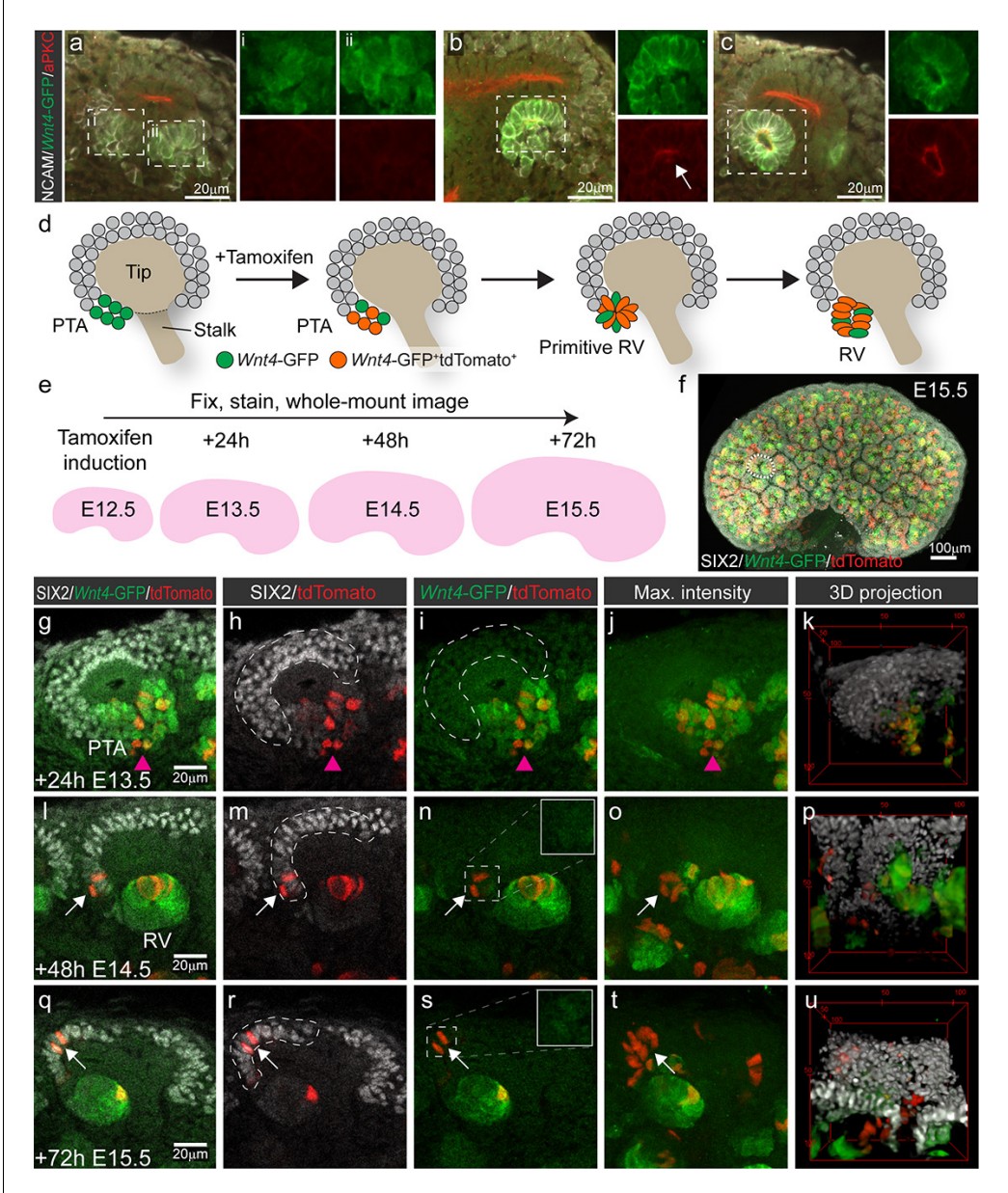

**Figure 1.** *Wnt4*-lineage cells re-enter the progenitor niche over time. (**a–c**) Correlation of *Wnt4*-GFP expression to stages of early nephron formation defined by Yang et al (*Yang et al., 2013*). Staining with NCAM (white, cell membrane), *Wnt4*-GFP (green), and aPKC (red) shows no polarization in *Wnt4*-expressing PTA structures in a, one aPKC-enriched foci in the primitive RV shown in b, and a polarised luminal epithelium within a mature RV in c. (**d**) Schematic of the experimental approach. In Wnt4$^{GCE}$;Rosa26R-LSL-tdTomato mice, committing cells within the PTA near the tip-stalk junction express GFP and become permanently labelled by tdTomato expression in the presence of tamoxifen. PTA structures mature to primitive RV and RV structures as nephrogenesis proceeds. (**e**) Cells were labelled with a single dose of tamoxifen at E12.5, then fixed and stained at E13.5, E14.5 or E15.5. Embryonic kidneys were stained for tdTomato protein (red), GFP (green) and nephron progenitor marker SIX2 (white) and whole-mount imaged. (**f**) Maximum intensity projection of a typical E15.5 dataset is shown, with a single niche marked by a dotted region. This image has been displayed on a black background for presentation purposes. (**g–k**) At 24 hr after tamoxifen (E13.5) tdTomato-labelled cells are restricted to the PTA (magenta arrowhead), marked by GFP expression and are absent from the cap mesenchyme marked by high SIX2 expression (dashed outline). (**l–p**) By 48 hr after tamoxifen (E14.5) rare tdTomato-labelled Wnt4 lineage cells (arrows) are found in the SIX2+ cap population (dashed outline). (**q-u**) At 72 hr after tamoxifen (E15.5) clusters of SIX2+ tdTomato-labelled cells (arrows) are found in some caps (dashed outline). Maximum intensity projections

*Figure 1 continued on next page*

*Figure 1 continued*
and 3D projections of confocal volumes show a cluster of cells present in a cap at E14.5 and E15.5 (arrows). Scale for all panels as in g, l, q, except for 3D projection where scale is indicated in bounding box.
DOI: https://doi.org/10.7554/eLife.41156.002
The following figure supplement is available for figure 1:

**Figure supplement 1.** Stromal Wnt4-lineage cells do not contribute to the nephron progenitor population.
DOI: https://doi.org/10.7554/eLife.41156.003

## Niche re-entry occurs throughout development

Nephrogenesis continues throughout development whilst the nephrogenic niche undergoes substantial changes in size (*Lindström et al., 2018b*; *Short et al., 2014*). These changes may affect the capacity of induced nephron progenitor cells to return to the cap mesenchyme. Lineage tracing *Wnt4*-expressing cells at E14.5, when the niches are more compact, labelled cells that were restricted to the PTA and RV after 24 hr, but gave rise to labelled SIX2$^+$ cells after 48 and 72 hr (*Figure 2b*). Thus, re-entry of *Wnt4*-lineage cells does not appear to be restricted to early kidney development.

We confirmed that tdTomato labelling within the nephron lineage was restricted to sites of nephron commitment throughout development by examining embryos 24 hr after injection at E12.5, E14.5 and E16.5 (*Figure 2—figure supplement 1*). *Wnt4*-GFP expression was not detected outside the tip-stalk junction at any stage. Thus, tdTomato-labelled nephron progenitor cells must be derived from cells within the site of nephron formation that have re-entered the nephron progenitor pool over time. This is supported by analysis of SIX2 and *Wnt4*-GFP expression within tdTomato +cells in our imaging data, which revealed a shift from a SIX2$^+$*Wnt4*-GFP$^+$tdTomato$^+$ state (cells were located at the tip-stalk junction), to a SIX2$^+$tdTomato$^+$ population over time (*Figure 2c*, *Figure 2—figure supplement 2*).

## Niche re-entry occurs from the onset of kidney development

Tamoxifen induced labelling of cells involves a delay of ~12–24 hr while the reporter mRNA is expressed and translated into protein. In order to investigate whether niche re-entry occurs during the first round of nephron induction, we labelled *Wnt4*-expressing cells at E10.5 and monitored labelled cells up to 96 hr after. Labelled cells expressing both SIX2 and *Wnt4*-GFP were observed around the base of the ureteric bud at E11.5. The majority of these cells contributed to nephrons while some labelled cells reintegrated into the cap mesenchyme (*Figure 2—figure supplement 3*). Some tdTomato-labelled cells remained within *Wnt4*-GFP$^+$ PTA-like regions 72 hr after labelling (*Figure 2—figure supplement 4*). Hence, while some cells may initiate *Wnt4* expression and then subsequently re-enter the cap mesenchyme, others remain at sites of induction without immediately progressing to epithelialization. In contrast, by E17.5, labelled cells within or near early committing structures appear to be restricted to niches with highly labelled nephron progenitor populations, suggesting that these cells arise from new inductions of previously labelled cells (*Figure 2—figure supplement 4*). Thus, while cells that remain induced may not immediately progress to a more mature nephron stage, such cells appear to ultimately commit.

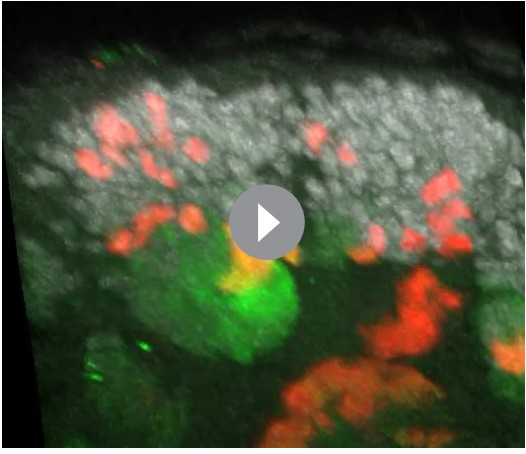

**Video 1.** 3D render of labelled cap cells in an E15.5 kidney after labelling at E12.5. A single niche has been manually segmented for display. Colours are SIX2 (grey), *Wnt4*-GFP (Green) and tdTomato (Red)
DOI: https://doi.org/10.7554/eLife.41156.004

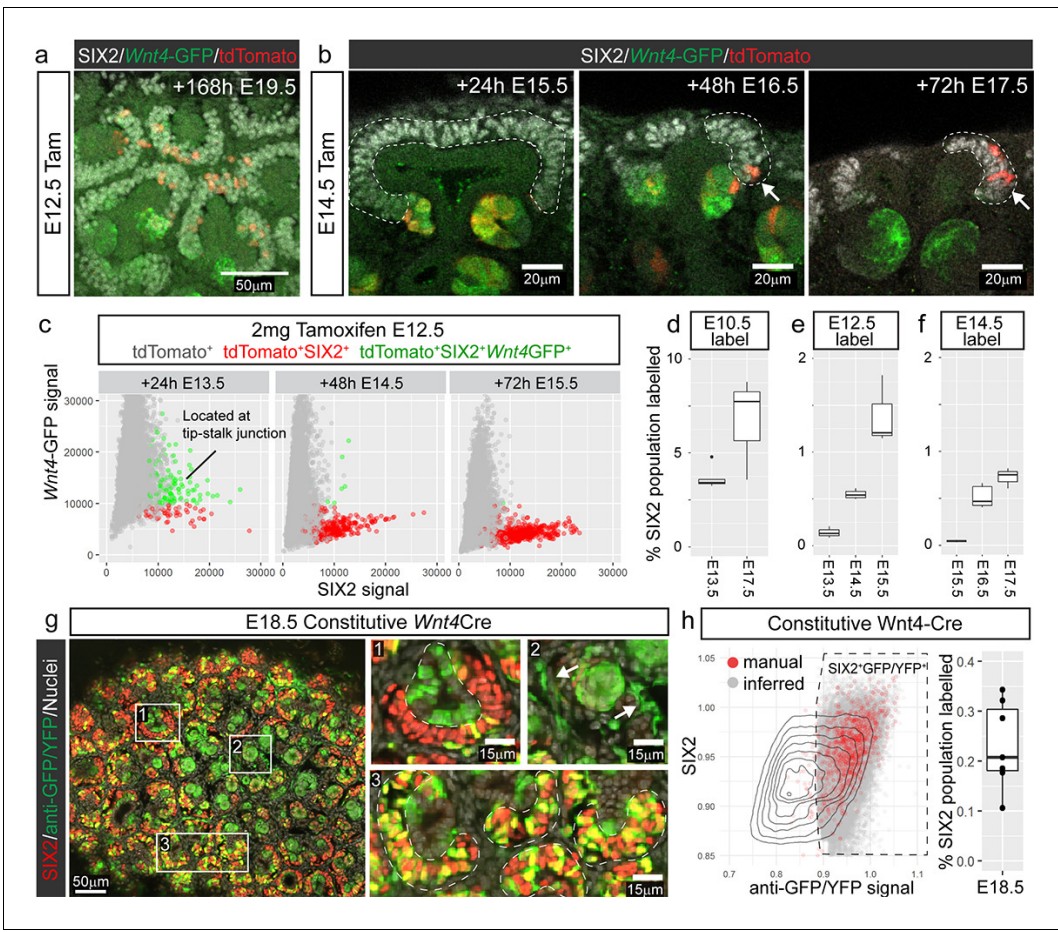

**Figure 2.** Cap re-entry occurs throughout development. (**a**) *Wnt4*-expressing cells at E12.5 can return to the cap mesenchyme and remain there for up to 7 days. (**b**) Labelling at E14.5 reveals cap re-entry later in development, with labelled cells absent from the SIX2 positive cap domain at E15.5, but a small number of cells present at E16.5 and E17.5. (**c**) Plot of SIX2 intensity vs *Wnt4*-GFP intensity for tdTomato-labelled cells identified by spot detection from wholemount imaging data. All dots represent tdTomato-expressing cells. Red cells are those assigned nephron progenitor (NP) identity, while green cells express *Wnt4*-GFP. The *Wnt4*-GFP cells at E13.5 that express high levels of SIX2 are located within PTAs. By E15.5 the tdTomato population includes a population of SIX2$^+$GFP$^-$ NP cells. Examples of wholemount data and spot classification are given in *Figure 2—figure supplement 2*. (**d–f**) Proportion of labelled SIX2$^+$ NP cells after tamoxifen treatment at E10.5, E12.5 and E14.5, determined by image analysis (see methods). (**g**) Labelling using a constitutive *Wnt4*-Cre gives rise to many positive cells in the SIX2$^+$ progenitor population at E18.5. A single confocal slice from a whole mount sample shows SIX2 (red) combined with anti-GFP staining (green) and nuclei (white). The anti-GFP antibody detects both *Wnt4* lineage (YFP$^+$) and present expression (*Wnt4*-GFP-Cre$^+$). Boxes 1-3 highlight less frequent labelling in the ureteric epithelium (1) and stroma (2), and extensive labelling of some caps (3). (**h**) ~ 20–30% of nephron progenitors are labelled by the constitutive *Wnt4*-Cre. Plot shows normalised nuclear spot-detection values, classified by either manual annotation (red) or inferred based on signal intensity (grey). Density lines indicate the entire SIX2$^+$ population, grey and red dots are SIX2$^+$ nuclei that are also expressing GFP and/or YFP. Signal thresholds were inferred per sample and used to estimate the proportion of labelled SIX2$^+$ cells.

DOI: https://doi.org/10.7554/eLife.41156.005

The following figure supplements are available for figure 2:

**Figure supplement 1.** Labelling within the nephron lineage is restricted to sites of nephron commitment throughout development.

DOI: https://doi.org/10.7554/eLife.41156.006

**Figure supplement 2.** Quantification of *Wnt4*-lineage cap cells.

DOI: https://doi.org/10.7554/eLife.41156.007

**Figure supplement 3.** *Wnt4*-expressing cells contribute to early nephrons and the nephron progenitor population at the onset of kidney development.

*Figure 2 continued on next page*

*Figure 2 continued*

DOI: https://doi.org/10.7554/eLife.41156.008

**Figure supplement 4.** Some induced cells may remain at sites of induction without immediately forming epithelial structures.

DOI: https://doi.org/10.7554/eLife.41156.009

## *Wnt4*-lineage cells accumulate, but remain a sub-population of the cap over time

Tamoxifen-mediated Cre activation in the *Wnt4*<sup>GCE</sup> line labels a proportion of *Wnt4*-expressing cells during a short window after delivery. While we observe an accumulation of *Wnt4*-lineage cells in the cap mesenchyme after a single labelling (*Figure 2d–f*), the extent of cells returning to the cap will be underestimated by this approach. In contrast, using a constitutive *Wnt4-GFP-Cre* line (*Mugford et al., 2009*) crossed to a *Rosa26-LSL-YFP* transgenic line (*Madisen et al., 2010*), facilitates study of the accumulation of returning cells across developmental time with the caveat that this may include unrelated events prior to kidney development. Frequent cap labelling (~20–30% of SIX2⁺ cells, *Figure 2h*) was observed at E18.5 using the constitutive *Wnt4*-Cre line, confirming that *Wnt4*-lineage 'escapers' accumulate in the cap mesenchyme throughout development, and ultimately represent a considerable proportion of the nephron progenitor population (*Figure 2g,h*). Nonetheless, a portion of the cap remained unlabelled suggesting that not all nephron progenitor cells arise from cells that have expressed *Wnt4* during the time period examined. In some cases we also observed labelling in the stroma and the ureteric tip (*Figure 2g*). As noted earlier, *Wnt4* is expressed at lower levels in some stromal cells, particularly in the medulla (*Georgas et al., 2008*; *Itäranta et al., 2006*). Stromal labelling was rarely observed in our timecourse experiments using the tamoxifen-dependant *Wnt4*<sup>GCE</sup>, suggesting that a higher level of tamoxifen is required to achieve a comparable level of labelling to the constitutive *Wnt4*-Cre.

## *Wnt4* lineage cells re-enter the cap by cell migration

To determine how tdTomato-labelled nephron progenitor cells come to be located in the cap mesenchyme we employed time lapse imaging of *Wnt4*<sup>GCE</sup>; *Rosa26-LSL-tdTomato*; *HoxB7-GFP* embryonic kidney explants. This enabled visualisation of *Wnt4* lineage cells around the ureteric tip as fluorescence from the *Wnt4*<sup>GCE</sup> transgene is weak in comparison to *Hoxb7*-GFP-marked tips. Kidneys were isolated at E12.5 and exposed to tamoxifen for an hour in culture before being washed. Samples were then grown for 18–24 hr prior to imaging to allow tdTomato fluorescence levels to reach a threshold required for detection and cell tracking (*Figure 3a,b*). Labelling equivalent samples at E12.5 and assessing at E13.5 in vivo resulted in tdTomato labelling of cells located at the tip-stalk junction in PTA-like structures prior to epithelialisation, and in more mature renal vesicles (*Figure 3c–e*). Imaging of cultured E12.5 kidneys fixed prior to live imaging showed a similar labelling pattern with tdTomato⁺ cells near the tip stalk junction in pre-epithelial cells and in primitive renal vesicles (*Figure 3f*). Live imaging E12.5 samples for 24 hr showed some *Wnt4*-lineage cells moving from the region under the tip to the top or end of a tip (*Figure 3g,j*, *Video 2*). Fixed analysis of these samples after live imaging shows that the tdTomato-labelled cells have migrated back into the cap mesenchyme and are surrounded by unlabelled SIX2-expressing cells (*Figure 3h,i,k,l*, *Figure 3—figure supplement 1*). The region under the tip that these cells migrated from contains a population of cells that have more prominent basolateral localisation of NCAM and lower levels of SIX2, suggesting those cells are undergoing the early stages of epithelialisation to form a nephron (*Figure 3h,k*). Using a tamoxifen inducible *Wnt4*-Cre in fixed samples across time, and by timelapse imaging, we show that *Wnt4* 'escapers' migrate out of sites of nephron induction and back into the progenitor niche.

The movement of nephron progenitor cells labelled using *Wnt4*<sup>GCE</sup> was compared to the movement of nephron progenitor cells labelled using *Six2*<sup>GCE</sup> to investigate whether prior expression of *Wnt4* influenced the movement of nephron progenitor cells (*Figure 3—figure supplement 2*). For each tiled dataset, movement of cells was calculated relative to their local niche by correcting for the movement of the adjacent GFP-marked epithelial tip (*Combes et al., 2016*; *Lefevre et al., 2016*). Comparing corrected migration speeds revealed that *Wnt4*-lineage cells displayed a small

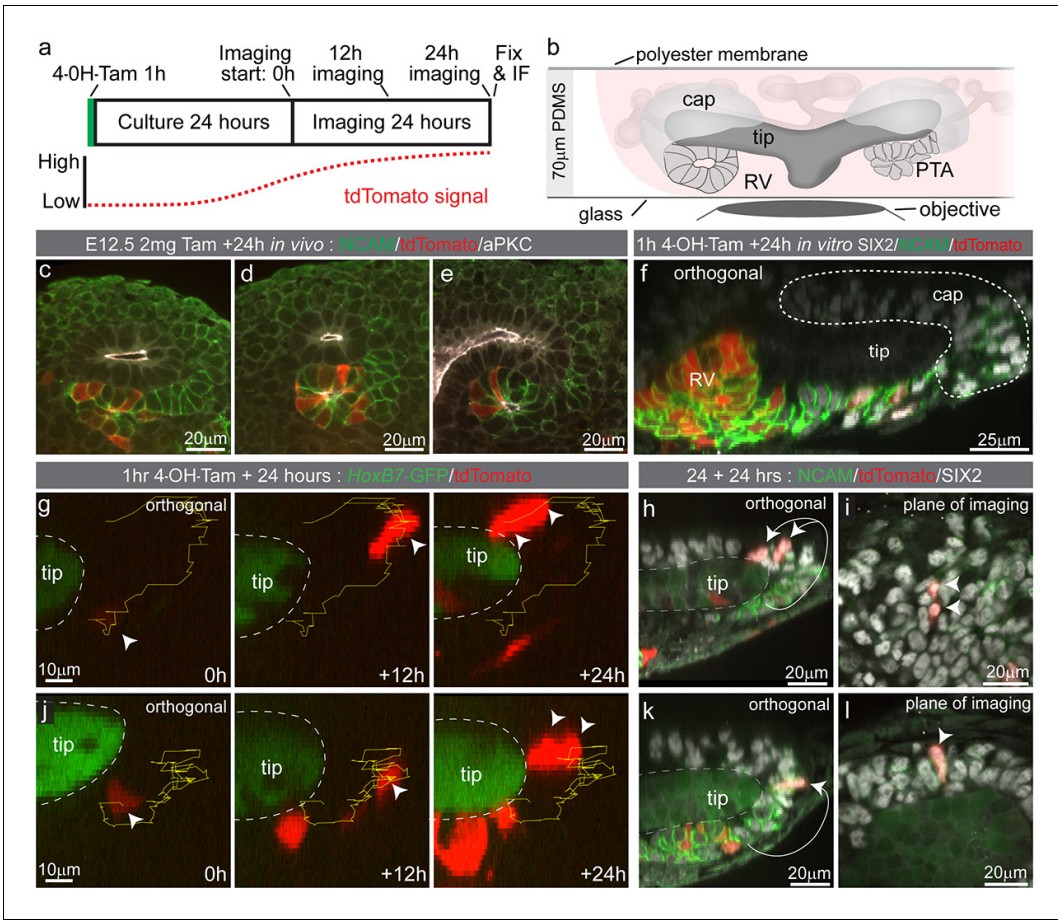

**Figure 3.** Some Wnt4 lineage cells migrate from sites of induction back into the cap mesenchyme. (a) Experimental overview for live imaging *Wnt4*-lineage cell migration. E12.5 embryonic kidneys were treated ex vivo for 1 hr (h) with 10 μM 4-Hydroxytamoxifen (4-OH-Tam), washed and cultured for 24 hr to allow tdTomato signal to increase to a level that could be detected. Samples were imaged for 24 hr at 15 min intervals, then fixed and immunostained. (b) Explant culture method. E12.5 kidneys were mounted in glass dishes in a chamber made from a 70 μm PDMS ring sealed at the top with a piece of porous polyester membrane. Explant cultures replicated in vivo niche geometry, allowing the entire niche to be captured as a high resolution Z-stack. (c–e) In vivo labelling 24 hr after tamoxifen administration at E12.5 is restricted to cells around the tip-stalk junction in PTAs and RVs where *Wnt4* is expressed. (f) 4-OH-Tam induction in explant culture replicates in vivo labelling. Labelling was observed in primitive nephron structures and at sites adjacent to the cap where *Wnt4* is expressed. The delay between tamoxifen exposure and the onset of imaging likely allows some movement of labelled cells prior to time point 0. View shows an orthogonal view along the Z axis (side view) of the 3D data. (g–i) and (j–l) show representative examples of cells moving from the base of the ureteric tip back into the cap mesenchyme. (g,j) At the start of imaging cells are observed at sites where PTA formation occurs. Over time cells make movements away from sites of commitment, up into the cap region. Yellow track indicates path over time, arrow heads indicate cell position, dashed outline shows border of tip. In each case cells undergo a division, as indicated by dual arrow heads at the final time point. Orthogonal view along the Z axis (side view) of 3D data is shown. Scales for all panels shown in left image. (h,k) Immunofluorescence of samples fixed immediately after 24 hr of imaging. SIX2 staining confirms nephron progenitor identity in tracked cells. NCAM highlights morphology in primitive nephron structures where the tracked *Wnt4*-lineage cells originate. Tip border is shown with a dashed line, simplified path of migration shown by white arrows. Second cell in k is in a different plane of imaging. Orthogonal view along the Z axis of the 3D data is shown. (i,l) XY axis views of the SIX2 +*Wnt4* lineage cells shown in h and k. Migration events shown are representative of events observed in 7 out of 14 separate 300 × 300 μm imaging fields derived from three independent explant experiments.

DOI: https://doi.org/10.7554/eLife.41156.010

The following figure supplements are available for figure 3:

*Figure 3 continued on next page*

*Figure 3 continued*

**Figure supplement 1.** Niche morphology is maintained in live explant cultures allowing the observation of Wnt4 lineage cell migration into the progenitor domain.
DOI: https://doi.org/10.7554/eLife.41156.011
**Figure supplement 2.** Comparison of cell movement between *Six2-* and *Wnt4*-lineage nephron progenitor cells.
DOI: https://doi.org/10.7554/eLife.41156.012

---

increase in mean migration speed compared to $Six2^{GCE}$ labelled progenitors (p<0.05, mixed effects model). This may reflect a requirement for larger movements to achieve cap re-entry.

## Modelling supports a process of positionally triggered stochastic induction

Nephron progenitor cells behave as a swarm, moving randomly within their local environment but maintaining their position relative to the ureteric tips as they grow and branch via attachment to the tip, and inferred attraction and repulsion cues (*Combes et al., 2016*). Our lineage tracing studies suggest that nephron commitment is not a unidirectional process. Some cells are induced to express *Wnt4*, but then reintegrate into the cap mesenchyme. Why then do some cells commit, while others do not? Though the precise events that dictate individual progenitor fate are not known, it is clear that regionally restricted cues trigger commitment at the site of nephron formation. One possibility is that commitment is dependent on the duration of exposure to inductive cues. To investigate this scenario further, we developed a computer simulation of nephron progenitor commitment in a motile, proliferating population (*Figure 4a,b*, Methods). In this model, cells are represented as points that migrate randomly while responding to attraction and repulsion cues from a simulated three-dimensional tip surface. Any cell entering a spatially restricted 'induction zone' in the lower region of the tip receives an inductive signal proportional to the time spent in the zone. Cells that reach a threshold level of inductive signal 'commit' and stop migrating, emulating an epithelialization event (*Figure 4b*). These parameters give rise to uncommitted cells at the top of the tip with committed progenitors accumulating within the induction zone at the bottom, mimicking nephron formation in vivo (*Figure 4c–f*, *Video 3*).

We simulated our lineage tracing experiments in silico by marking cells that entered the inductive zone at a defined time (*Figure 4d,f*). This allowed a comparison between the predictions of the model and experimental labelling with the tamoxifen-dependent *Wnt4*-Cre. The model predicted that labelling progenitor cells as they received an inductive signal gave rise to committed cells as well as cells that moved back out of the induction zone before reaching the threshold for commitment (*Figure 4c,d*). Thus, the in silico model supports our experimental finding that *Wnt4*-Cre labels both committing cells and a proportion of the nephron progenitor population.

Our initial model allowed induced progenitors that re-enter the progenitor pool to retain their state and remain 'primed', increasing the chance of future commitment (*Figure 4c,d*). In this scenario, as more cells accumulate inductive events the proportion of uninduced progenitors decreases with time (*Figure 4g*). We developed an alternative model that allowed induced cells to return to their original uninduced 'ground state' in response to time spent back in the progenitor pool (*Figure 4e,f*). This scenario gave rise to a more sharply defined domain of cells that have recently undergone induction, where the proportion of uncommitted cells remained steady for a given niche size (*Figure 4e,f,g*).

The size of the nephron progenitor niche decreases across development in mouse and human (*Lindström et al., 2018b*; *Short et al., 2014*). In both simulations, decreasing niche size or increasing the rate of random cell movement resulted in an increased proportion of induced cells because cells were more likely to move into the induction zone (*Figure 4g,h*). In the primed scenario the proportion of uninduced

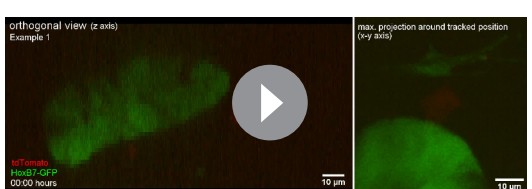

**Video 2.** Some Wnt4 lineage cells migrate from sites of induction back in the cap mesenchyme. Example one corresponds to panels in *Figure 3g–i* and Example two to *Figure 3j–l*.
DOI: https://doi.org/10.7554/eLife.41156.013

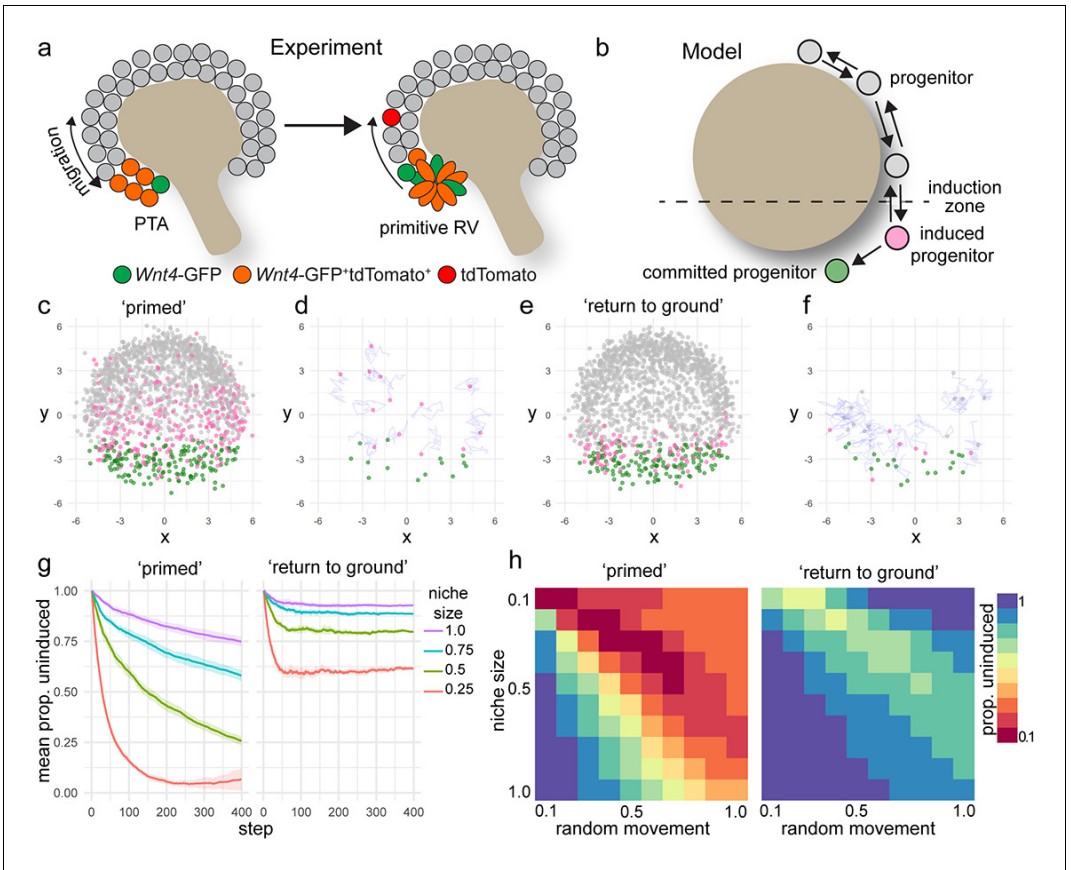

**Figure 4.** A model of positionally triggered stochastic induction. (**a**) Proposed mechanism by which *Wnt4* lineage cells re-enter the niche. Migrating cells enter the induction zone where *Wnt4*-expression/PTA initiation occurs. A portion of cells make sufficient movements to 'escape' and re-enter the cap domain, while those remaining form a primitive RV. (**b**) Diagram of simulation: randomly migrating progenitor cells (grey) become induced (pink) in response to regionalized cues and commit (green) after accumulation of sufficient inductive cues. (**c–f**) Simulated migrating cells begin in the progenitor state (grey) and give rise to induced (pink) and committed cells (green). (**c, d**) Simulations at step 200. (**d,f**) Fate of cells that were undergoing induction at step 100, shown at step 200. Many cells have committed, while a portion re-enter the cap and continue to migrate. Tracks show recent movements. (**g**) Proportion of uninduced cells over time from 'primed' and 'return to ground' models. Mean and standard deviation for 10 simulations are shown. (**h**) Proportion of uninduced cells at simulation step 400 (mean of 10 simulations) for changes in niche size and random migration for each variant of the model.
DOI: https://doi.org/10.7554/eLife.41156.014

progenitors dropped sharply as niche size decreased, or migration rate increased leading to a collapse of the uncommitted population (*Figure 4g,h*). Changing niche size or migration rate in the 'return to ground' model resulted in a stepped decrease in the proportion of uninduced progenitors but reached equilibrium due to the capacity of induced progenitors to revert to an uninduced state (*Figure 4g,h*).

These models highlight a mechanism by which randomly migrating cells can give rise to a spatially restricted population of committed cells that is responsive to changes in the niche. Both models result in regionalized enrichment of uncommitted and induced progenitors. Thus, at a population level, our models are broadly consistent with the established domain-based understanding of the cap mesenchyme

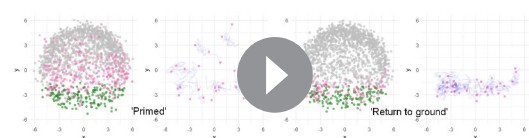

**Video 3.** Animation of computational simulation of stochastic commitment.
DOI: https://doi.org/10.7554/eLife.41156.015

(*Brown et al., 2013*; *Mugford et al., 2009*). In both the original 'primed' scenario and the 'return to ground' scenario, cells that were labelled as they received an inductive cue gave rise to both committed cells and 'escapers' in the progenitor pool (*Figure 4c–f*). However, the underlying state of the 'escaper' population differs between these two models with these differences predicting changes in the capacity of the niche to maintain an uncommitted population.

## *Wnt4* labelled nephron progenitor cells exist in a range of transcriptional states

To test whether nephron progenitor cells that had expressed *Wnt4* are 'primed' or revert to a 'ground' state we used single cell RNA sequencing (scRNA-seq) to interrogate the transcriptional profile of *Wnt4*-lineage nephron progenitor cells vs unlabelled nephron progenitor cells. To maximise labelling we gave four daily doses of tamoxifen from E10.5 to E13.5 and dissected at E15.5. Labelling occurred at a higher frequency than for a single dose of tamoxifen (*Figure 5a*, *Figure 5—figure supplement 1*). Labelled E15.5 kidneys were dissociated to generate libraries for single cell RNA sequencing using the 10x Chromium platform. Surplus cells were analysed on a flow cytometer to confirm tdTomato expression and cell viability (*Figure 5—figure supplement 1*).

We used the Seurat library (*Butler et al., 2018*) to perform unsupervised clustering on a filtered dataset of 3451 cells and identified 10 main transcriptional clusters corresponding to nephron progenitor (marked by *Six2*) and committed nephron cells (two clusters, *Pax2*, *Pax8*), vasculature (*Pecam1*), stroma (five clusters, *Pdgfra*) immune cells (*Tyrobp*), and erythrocytes (*Hbb*) (*Figure 5—figure supplement 2*). Lists of whole kidney cluster marker genes are provided in *Supplementary File 1*. Focussing on the 604 cells that made up the nephron progenitor and epithelial clusters, we performed clustering at a higher resolution to identify distinct sub-populations, revealing populations that correspond to nephron progenitor cells (two clusters, *Six2, Cited1,* with cluster separation based on a weak stromal signature e.g. *Pdgfra*), committing (PTA and RV, *Wnt4*) cells, differentiated tubule cells (*Pax8*), podocytes (*Mafb*) and ureteric epithelium (UE, *Wnt9b*) (*Figure 5b,c*, *Figure 5—figure supplement 3*). Lists of nephron cluster marker genes are provided in *Supplementary File 2*. To identify lineage-labelled cells, we mapped reads corresponding to the Rosa26-tdTomato transcript, present only in *Wnt4*$^{GCE}$ lineage-labelled cells that have undergone a Cre-mediated recombination event to remove the preceding stop codon. Labelled cells were present within committing and tubule cells and podocytes, as expected (*Figure 5b,c*). We also detected tdTomato-labelled cells within nephron progenitor clusters (*Figure 5c,d*). 27/42 labelled cells expressed both *Six2* and *tdTomato*, but not *Wnt4* suggesting that they represent the *Wnt4* labelled nephron progenitor (escaper) population (*Figure 5c,d*). While some cells within the nephron progenitor cluster expressed low levels of *Wnt4* RNA, we only observed co-expression of SIX2 and *Wnt4*-GFP protein in PTA and RV cells (*Figure 1*, *Figure 2—figure supplement 1*); as such, we interpret cells co-expressing *Wnt4* and nephron progenitor markers such as *Cited1* as cells that are transitioning into or out of a committing state. There was no evidence for significant differential expression between labelled cells and unlabelled cells within progenitor clusters (average log fold change threshold 0.25). The existence of *Wnt4* lineage cells with a transcriptional profile identical to surrounding nephron progenitors most closely supports our 'return to ground' state computational model, whereby cap re-entry allows cells to return to an uninduced progenitor state. This is also consistent with *Wnt4*-labelled cells remaining in the cap mesenchyme for up to 7 days (*Figure 2a*). Similarly, *tdTomato* expressing cells were found within all nephron progenitor subpopulations, suggesting that previous expression of *Wnt4* does not give rise to a distinct nephron progenitor state.

Genes differentially expressed between nephron progenitor and committed nephron cells included established nephron progenitor markers such as *Cited1* and *Six2*, and differentiation markers such as *Pax8*, *Jag1* and *Lhx1* (*Figure 5d*, *Figure 5—figure supplement 3*). Within the *Wnt4*-lineage-labelled nephron progenitor population, there was transcriptional heterogeneity. Some cells showed a strong progenitor profile, while others appeared in a partially induced state with co-expression of genes previously regarded as delineating progenitor and committed states (*Figure 5d*). This observation is in agreement with the previously noted transcriptional plasticity of the nephron progenitor population (*Brunskill et al., 2014*).

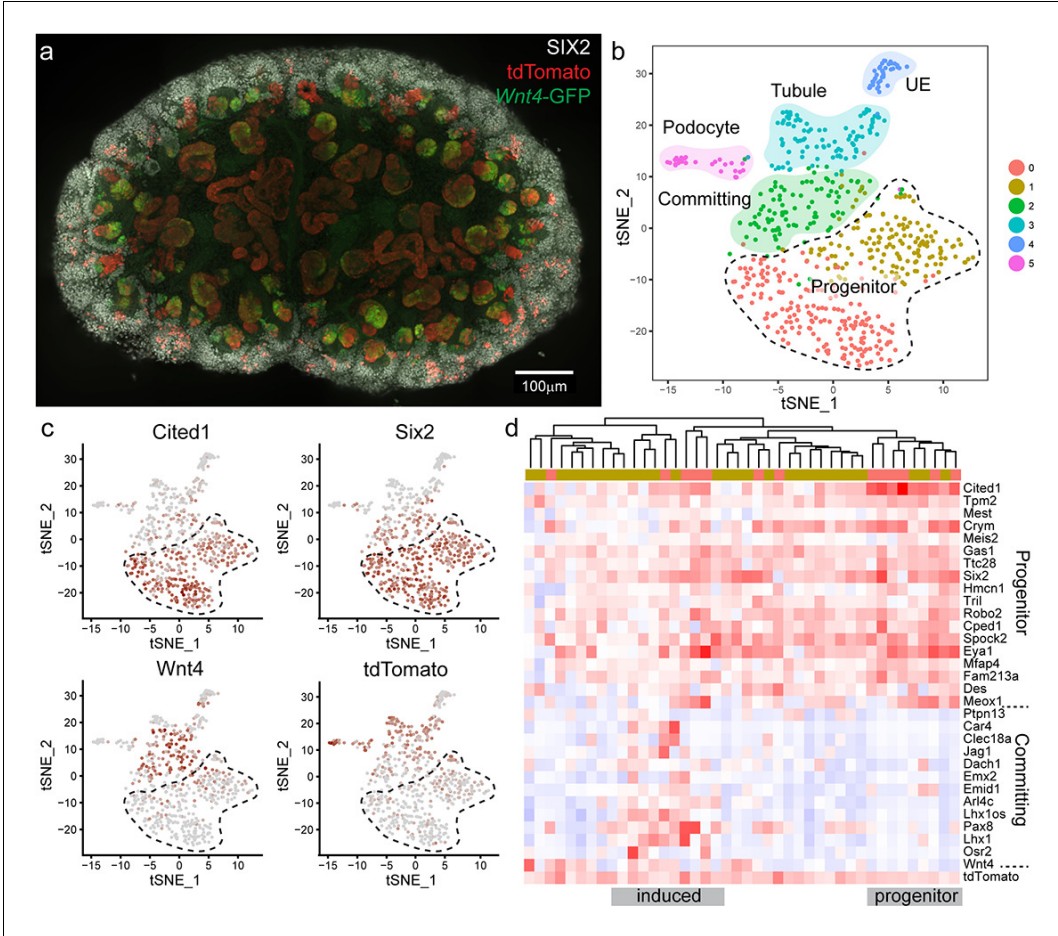

**Figure 5.** *Wnt4* labelled progenitors accumulate in the cap and show a range of transcriptional identities. (**a**) E15.5 kidney after daily tamoxifen injection at E10.5, 11.5, 12.5 and 13.5, showing increased labelling within the cap population (grey, SIX2; red, tdTomato; green, *Wnt4*-GFP). A maximum projection of a subset of a wholemount confocal stack is shown. (**b**) tSNE plot representing clustering of single cell transcriptional profiles. Clusters represent nephron progenitors (two clusters), committing cells (PTA and RV), differentiated tubule, podocyte and ureteric epithelium (UE). (**c**) tSNE plots coloured by scaled gene expression show nephron progenitor (NP) markers *Six2* and *Cited1*, committing cell marker *Wnt4* and *tdTomato* expression. tdTomato-labelled cells are present within both the NP and committing cells. (**d**) Heatmap showing expression of markers associated with the shift from progenitor to committed state for tdTomato expressing NP cells (cells from nephron progenitor cluster 0 and 1). Dendrogram based on this subset of genes reveals labelled NP cells in both a strong progenitor-like state with high progenitor gene expression and weak committing gene expression (including absence of *Wnt4*), and a partially induced-progenitor state with co-expression of both marker types. Remaining cells show a progenitor profile with heterogeneous expression of *Wnt4*, suggesting that they may be transitioning into or out of the induced state. Column colour labels indicate cluster identity, as in (**b**).

DOI: https://doi.org/10.7554/eLife.41156.016

The following figure supplements are available for figure 5:

**Figure supplement 1.** Daily tamoxifen injections give strong labelling in the cap.
DOI: https://doi.org/10.7554/eLife.41156.017

**Figure supplement 2.** Whole kidney clusters and marker expression.
DOI: https://doi.org/10.7554/eLife.41156.018

**Figure supplement 3.** Segment markers and expression changes with commitment.
DOI: https://doi.org/10.7554/eLife.41156.019

## Induced progenitors make a delayed contribution to multiple nephrons

Our model predicts that induced cells may remain motile for varying periods of time as they accumulate sufficient cues to eventually epithelialise. Transcriptional profiling supports this possibility, revealing partially committed cells that co-express progenitor markers (*Figure 5e*). To attempt to view such cells we focused our live imaging on *Wnt4*-expressing cells in the PTA of E11.5 explants that have yet to epithelialise. Labelling was induced with a short 35 min pulse of 4-OH-tamoxifen in culture, which was washed out before culture prior to live imaging. Under these conditions many labelled cells remained within the zone of early nephron formation rather than returning to the cap mesenchyme, but did not immediately form a renal vesicle.

Clusters of *Wnt4*-lineage cells at sites of nephron induction were imaged in E11.5 explants over 44 hr. Cells from these initial aggregates were motile and contributed to several adjacent nephrons, counter to the established understanding that a single induction event gives rise to a single nephron (*Figure 6a*, *Video 4*). At the completion of live imaging explants were fixed and stained for JAG1, a marker of early nephrons, revealing that *Wnt4*-expressing cells labelled at E11.5 contributed to the majority of nephrons over time (*Figure 6a*). By comparing high resolution fixed images with live imaging data, we assigned committed cells in developing nephrons back to the *Wnt4*-labelled progenitors from which they were derived (*Figure 6b–k*). Groups of *Wnt4* lineage cells that were within 1–3 cell diameters of each other at the onset of imaging ultimately contributed to multiple adjacent nephrons (*Figure 6b–k*). Thus progenitors in the early developing kidney are able to commit via a delayed process whereby a group of motile induced progenitors can give rise to multiple nephrons over the proceeding branching iterations. The fact that such cells have both initiated *Wnt4* expression and remain in a region in which they should be able to receive a positional inductive event implies that additional events downstream of WNT9B secretion and *Wnt4* expression are required for nephron initiation. Once again, the inclusion of cell movement into our existing paradigm of nephron initiation challenges our understanding of how this process is regulated. It is currently unclear whether induced progenitors undergo delayed commitment or contribute to more than one nephron at later stages of kidney development when niches are smaller and more discrete.

## Discussion

The formation of sufficient nephrons within the mammalian kidney is thought to depend on a balance between nephron progenitor expansion and commitment. How this balance is regulated to ensure optimal organ growth and eventual cessation is unresolved. Our previous observations that nephron progenitors behave as a motile swarm, responding to contradictory signals from the adjacent tip, were difficult to reconcile with the established view of a linear, sequential process of commitment. In this study, we have used *Wnt4* lineage tracing to investigate the process of nephron commitment at higher resolution. Contrary to expectation, not all cells that enter the zone of nephron formation commit.

Based on timelapse imaging, computational modelling, and scRNA-seq, we propose a refined model of nephron progenitor commitment in which cell fate is dependent on the duration of exposure to spatially-restricted inductive cues, with this being influenced by random cell migration. This model predicts that *Wnt4* expressing cells in the lower cap are in a plastic state. Depending on the duration of exposure they may either commit to forming a nephron or re-enter the progenitor pool. Under this model, a population of swarming progenitors gives rise to a constant pool of committing cells. This reconciles the dynamic behaviour that we have previously reported in the developing kidney (*Combes et al., 2016*) with the established notion of spatially restricted inductive cues.

Our data support that *Wnt4* expression immediately precedes the formation of a committed aggregate. Though we observe considerable motility in *Wnt4*-lineage-labelled cells we cannot distinguish between motile cells at the tip-stalk junction that have initiated *Wnt4* expression but have yet to form an aggregate, and those that may have migrated out of a partially aggregated group of cells. Given that most cells within the PTA region go on to form nephrons we predict that the ability to undergo cap re-entry is quickly lost as cells aggregate. Thus, re-entry events may be restricted to cells that are excluded as an aggregate forms. This is consistent with our model, whereby the fate of progenitors in the tip-stalk junction depends on migration events that occur immediately prior to the switch to a non-motile state. The events that define the point at which cap re-entry is no longer

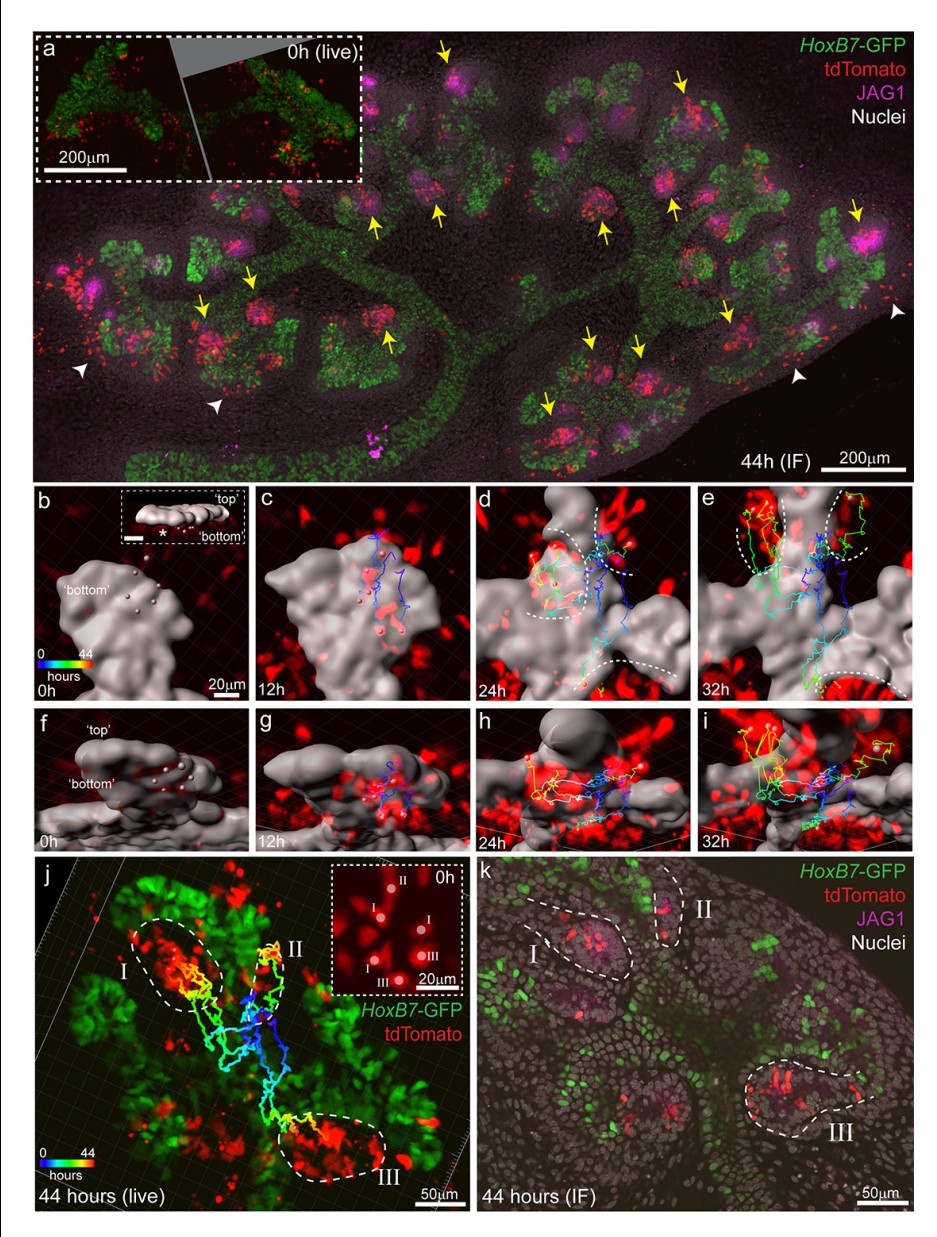

**Figure 6.** Induced progenitors make a delayed contribution to multiple nephrons at E11.5. (a) Representative example of cultured explants from *Wnt4*$^{GCE}$; Rosa26-LSL-tdTomato; *HoxB7*-GFP E11.5 embryos. Inset of live imaging at time point 0 hr (h) (~24 hr after 35 min of 4-Hydroxytamoxifen induction) shows groups of *Wnt4*$^{GCE}$-labelled cells clustered around sites where early nephrons are being induced to form under the tip. The same explant was fixed and stained for developing nephron marker JAG1 at the completion of 44 hr of live imaging. Labelled cells are present within many nephrons (yellow arrows), inconsistent with a model of a single induction event giving rise to a single nephron. A portion of cells are also present outside of nephrons in the cap mesenchyme region (white arrowheads). (b–i) An example of a group of 6 tdTomato-labelled cells migrating to contribute to three nephrons around adjacent tips. *HoxB7*-GFP tip signal is rendered as a grey surface. Images in b-e are oriented to show a view of the 'bottom' side of the tip where nephrons form. (b) 3D render shows the starting positions of tracked cells restricted to a region below a single tip. Inset in (b) shows a side view indicating the position of cells below the tip, inset scale is 20 µm. Cell centres represented as grey spheres with tracking indicated by lines coloured to indicate time. (c–e) By 24 hr tracked cells are evident in three regions where separate nephrons are forming (dotted lines). (f–i) Rotated view of b-e showing tracks under the tip. (j) 3D render of movie end-point, with tracking indicated. After 44 hr tracked cells are evident in three distinct nephrons (dotted regions I, II, III). Inset shows a maximum projection around the initial group of cells with labels indicating the

*Figure 6 continued on next page*

*Figure 6 continued*
nephron that each cell eventually contributes to. In this example cells within 1–3 cell widths of each other contribute to distinct nephrons. (k) Immunofluorescence at the completion of imaging confirms that final tracked positions correspond to three distinct nephrons (JAG1 stain, dotted regions) that contain labelled cells. Example shown is representative of events observed in five 614 × 614 µm fields captured from three kidneys explants.
DOI: https://doi.org/10.7554/eLife.41156.020

possible are not well understood and may involve a gradual increase in adhesion, or a sharp switch from motile individual cells to an aggregated state.

The WNT9B-mediated induction of *Wnt4* has been regarded as essential for commitment of SIX2[+]nephron progenitor cells to nephron formation for more than a decade (*Carroll et al., 2005*; *Kispert et al., 1998*; *Park et al., 2012*; *Park et al., 2007*; *Stark et al., 1994*). *Wnt9b* expression is seen within the ureteric tip and while both nephron progenitor turnover and induction is regarded as a WNT9B-mediated canonical Wnt signalling event (*Karner et al., 2011*; *Ramalingam et al., 2018*), the receptor complexes involved are not defined. Phosphorylated SMAD1/5-mediated BMP signalling has been identified as a factor required to transition nephron progenitor cells into a state where they are receptive to inductive cues (*Brown et al., 2013*). Recent studies would suggest a capacity for Notch activation to induce nephron commitment (*Chung et al., 2016*; *Chung et al., 2017*), consistent with the commitment of all nephron progenitors after targeted activation of Notch signalling within the cap mesenchyme (*Boyle et al., 2011*). A number of other observations also suggest that the spatially regulated positioning of nephron formation may involve more than one initiating signal. These include subtle differences in phenotype between *Fgf8* and *Wnt4* mutant kidneys (*Perantoni et al., 2005*; *Stark et al., 1994*) and evidence for stromal-derived signals regulating commitment (*Das et al., 2013*; *Fetting et al., 2014*). Our observation of cells which receive a trigger to initiate *Wnt4* expression but fail to commit may indicate a requirement for prolonged induction or a second, as yet undefined, consolidating trigger. The point at which this cascade of events passes a 'point of no return' is unknown, though the physical constraint of epithelialisation likely prevents return to the progenitor population. These data predict that lineage tracing using a later marker of commitment (eg *Jag1*, *Lhx1*) would not give rise to cells that re-enter the nephron progenitor population.

Single cell transcriptional profiling revealed that *Wnt4* lineage cells that re-enter the cap mesenchyme become indistinguishable from the remaining progenitor pool. Although the absence of a unique transcriptional profile for the *Wnt4* labelled nephron progenitor 'escapers' may be due to a lack of transcriptional depth in the single cell approach, there was sufficient depth to place these cells within cap mesenchyme clusters. Previous heterochronic transplantation studies have suggested that there is a transcriptional signature related to cap mesenchyme 'aging' but that placing an 'old' nephron progenitor back within a 'young' niche extended the time across which this cell would remain a nephron progenitor (*Chen et al., 2015*). We found no evidence for differential expression of markers associated with nephron progenitor aging in *Wnt4*-Cre-labelled vs unlabelled progenitors. However, our data are consistent with the proposed influence of surrounding cells on the transcriptional state of any individual nephron progenitor cell within the niche. In our 'return to ground state' model, individual nephron progenitor cell fate depends on migration events that modulate exposure to niche cues, yet the nephron progenitor population as a whole reaches an equilibrium for a given set of values. Thus, in our paradigm, cells from an older population with a greater proportion of induced cells may adjust to match the equilibrium of a 'young' niche state given appropriate instructive cues.

Computational simulations of our stochastic model are consistent with eventual cessation of nephrogenesis. We propose that the duration of inductive signal required for commitment could remain steady, with a reduction in niche size

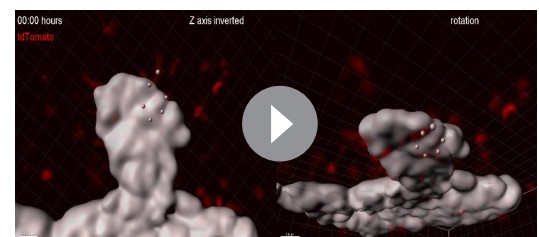

**Video 4.** Induced progenitors make a delayed contribution to multiple nephrons at E11.5.
DOI: https://doi.org/10.7554/eLife.41156.021

shifting the equilibrium toward a more primed population as cells spend more time in zones of induction and less time in receipt of tip signals promoting progenitor maintenance. This mechanism may provide a means to couple cessation of nephron formation with organ growth, invoking only a relatively simple set of niche behaviours. Observations that cap mesenchyme cell cycle length, and hence proliferation rate, reduces with developmental time (*Short et al., 2014*) and that cap populations represent a more committed state with time (*Chen et al., 2015*) are consistent with this model of stochastic commitment in a shrinking niche. Indeed, we have also previously shown in mouse that around birth there is a simultaneous induction of nephrons all the way around the terminal ureteric tips with all remaining surrounding cells forming RVs (*Rumballe et al., 2011*).

In conclusion, this study provides an alternate model to explain the balance between nephron progenitor maintenance and nephron commitment. This model is consistent with both the conventional anatomical positioning of nephron formation and our documented evidence of a motile cap mesenchyme population. It also implies a requirement for a prolonged or additional signal to nephron progenitors to ensure commitment to epithelialisation as well as predicting that stochastic commitment influenced by cell migration can explain cap mesenchyme exhaustion. This revised view of how nephron formation and cessation is controlled may have implications for understanding nephron deficit observed in response to both genetic and environmental cues.

# Materials and methods

## Key resources table

| Reagent type (species) or resource | Designation | Source or reference | Identifiers | Additional information |
|---|---|---|---|---|
| Genetic reagent (*Mus musculus*) | Wnt4$^{GCE}$; Wnt4-GFP-CreER$^{T2}$ | PMID: 18682239 | RRID:IMSR_JAX:032489 | |
| Genetic reagent (*Mus musculus*) | Rosa26-LSL-tdTomato | PMID: 20023653 | RRID:IMSR_JAX:007909 | |
| Genetic reagent (*Mus musculus*) | Hoxb7-GFP | PMID: 10322632 | RRID:IMSR_JAX:016251 | |
| Genetic reagent (*Mus musculus*) | Wnt4-GFP-Cre | PMID: 19591821 | RRID:IMSR_JAX:032490 | |
| Genetic reagent (*Mus musculus*) | Rosa26-LSL-YFP | PMID: 20023653 | RRID:IMSR_JAX:007903 | |
| Genetic reagent (*Mus musculus*) | Six2$^{GCE}$; Six2-GFP-CreER$^{T2}$ | PMID: 18682239 | RRID:IMSR_JAX:009600 | |
| Genetic reagent (*Mus musculus*) | PdgfraCreER | PMID: 23335233 | CDB0674K RIKEN Center for Life Science Technologies | More information |

## Mouse strains and tissue collection

Mouse experiments were conducted with approval of the Murdoch Children's Research Institute Animal Ethics Committee, or the Cincinnati Children's Hospital Medical Center (CCHMC) animal studies committee. *Wnt4* lineage tracing experiments were performed by mating *Wnt4*$^{GCE/+}$ (*Wnt4*tm2 (EGFP/cre/ERT2)Amc) (*Kobayashi et al., 2008*); Rosa26-LSL-tdTomato (Gt(ROSA)26Sortm14(CAG-tdTomato)Hze) (*Madisen et al., 2010*) males to C57BL/6 females. Migration analysis was performed by mating *Wnt4*$^{GCE/+}$; Ros26-LSL-tdTomato males to *HoxB7*-GFP (129S.Cg-Tg(*Hoxb7*-EGFP)33Cos/ J) (*Srinivas et al., 1999*) females, or *Six2*$^{GCE/+}$ (*Six2*tm3(EGFP/cre/ERT2)Amc) (*Kobayashi et al., 2008*); *HoxB7*-GFP males to Rosa26-LSL-tdTomato females. Constitutive *Wnt4*-Cre-labelled was performed by mating constitutive *Wnt4*-GFP-Cre (*Wnt4*tm3(EGFP/cre)Amc) (*Mugford et al., 2009*) mice to Rosa26-LSL-YFP (Gt(ROSA)26Sortm3(CAG-EYFP)Hze/J) (*Madisen et al., 2010*) mice. Inducible *Pdgfra* Cre mice (PDGFRa-MerCreMer, RIKEN Centre for Life Science Technologies CDB0674K) (*Ding et al., 2013*) were bred to Rosa26-LSL-tdTomato mice (*Madisen et al., 2010*). Timed matings were checked daily with E0.5 considered as noon of the day the seminal plug was identified. 2 mg tamoxifen (Sigma T5648) in corn oil (Sigma C8267) was administered to pregnant females by IP injection to initiate tamoxifen-inducible cre activity unless otherwise stated. Embryos were collected at the indicated times, dissected and fixed for 10–20 min in 4% PFA in PBS.

## Immunofluorescence

Samples were blocked at least overnight in 5% normal donkey serum, 5% normal goat serum in TBS + 0.1% Triton-X, or PBS + 0.1% Triton-X prior to treatment in primary antibody diluted in blocking solution. Washes and antibody incubations were performed in TBS + 0.1% Triton-X or PBS + 0.1% Triton-X at 4°C for at least 24 hr each step. Antibodies used were Rabbit anti-SIX2 (Proteintech, 11562–1-AP), Mouse anti-SIX2 (clone 3D7, Abnova, H00010736-M01), Rabbit anti-RFP (MBL, PM005), Chicken anti-GFP (Abcam, ab13970), Rabbit anti-JAG1 (Cell Signalling, 2620), Rabbit anti-aPKC-zeta (Santa Cruz, sc-216), Rat anti-NCAM (GeneTex, GTX19782). Secondary antibodies labelled with Alexa-488, Alexa-568, Alexa-647 or BV-421 were purchased from Thermo Fisher or Jackson ImmunoResearch. In some experiments nuclei were labelled with Draq5 (Abcam, ab108410). Samples for *Wnt4* lineage analysis were cleared using the previously described BABB based method (*Combes et al., 2014*). Constitutive *Wnt4*-Cre-labelled samples were cleared with Ethyl Cinnamate (*Klingberg et al., 2017*). Samples to identify native tdTomato signal alongside aPKC and NCAM were cleared using passive clarity (*Yang et al., 2014*). Samples were mounted in 35 mm glass bottom dishes (MatTek) for imaging.

## Whole mount microscopy and flow cytometry

Whole mount samples for lineage analysis were imaged using a Zeiss LSM780, with a Zeiss 0.8 NA 25x multi-immersion objective. Typical datasets consisted of tiled images at 0.5 x 0.5 x 2 µm voxel size. All other fixed datasets were imaged using an Andor Dragonfly spinning disk confocal system using a Nikon 40 × 1.15 NA water immersion objective. Datasets had a 0.3 x 0.3 x 0.5 µm voxel size (multi label to match single cell experiment) or 0.3 x 0.3 x 1 µm voxel size (all others). Tiled images were stitched using the stitching plugin within Fiji (*Preibisch et al., 2009*; *Schindelin et al., 2012*). Large datasets were visualised using Imaris (Bitplane) or the Fiji BigDataViewer plugin (*Pietzsch et al., 2015*). Flow cytometry was performed using a X-20 Fortessa analyser (BD) with DAPI at 0.5 ug/ml as a viability dye.

## Live microscopy and explant culture

High resolution time lapse imaging was performed using a modified version of the 'Fixed-Z' method (*Saarela et al., 2017*). E12.5 explants were mounted in glass bottom six well plates (Cellvis) surrounded by a ring of 70 µm thick PDMS (Sylgard 184, Dowsil) with a piece of porous polyester transwell (0.4 µm pore size, Corning #3450) serving as a 'lid' so that the sample was sandwiched between transwell and glass (See *Figure 3—figure supplement 1*). E11.5 explants (*Figure 6*) were mounted using a similar approach but substituting the transwell membrane for a thin PDMS 'lid' with two 70 µm PDMS strips serving as spacers at either side. When assembled carefully to prevent wetting contact areas the PDMS spacers bound to the glass dish and to the PDMS or transwell 'lid' and remained in place for many days after the dish was flooded with media, restricting the sample to a 70 micron region above the coverslip.

*Wnt4*[GCE] labelling in E11.5 kidneys (*Figure 6*) was activated with 100 nM 4-Hydroxytamoxifen ex vivo for 35 min at room temperature. Wnt4[GCE] labelling in E12.5 kidneys (*Figure 3*) was activated with 10 µM 4-Hydroxytamoxifen (Sigma HS278) for 1 hr at room temperature. We find variability between batches of 4-Hydroxytamoxifen that necessitates optimisation of the dosage for each new batch. 4-Hydroxytamoxifen was stored at −30C in ethanol and diluted in in $CO_2$ independent media (Thermo-Fisher) or DMEM/F-12 + Hepes (Thermo Fisher) fresh before use. Explants were cultured in DMEM +10% Fetal calf serum, or phenol red free DMEM +10% Fetal calf serum with Penicillin-Streptomycin (Thermo Fisher).

Samples were imaged using an Andor Dragonfly spinning disk confocal with 1.15NA 40x long working distance water immersion objective (*Figure 3*) or a 0.75 NA 20x air objective (*Figure 6*). Z stacks were taken at 15 min intervals at 2 µm Z spacing. In some cases imaging was briefly paused to adjust stage position. Samples that were required for further IF analysis were immediately fixed in 4% PFA in PBS for 10 min at the completion of live imaging.

To compare migration rates *Wnt4*-Cre-labelled embryos were treated with 2 mg tamoxifen by IP injection to the pregnant mother at E11.5, dissected at E12.5 and mounted on transwells for live imaging following previous methods (*Combes et al., 2016*). Control Six2-Cre-labelled embryos were dissected at E12.5 and treated with 1 nM 4-Hydroxytamoxifen ex vivo for 35 min prior to mounting

on transwells. This concentration gave a very low induction amenable to single cell tracking. Live imaging was performed using a Zeiss LSM780 with 40 × 1.0 NA long working distance water immersion objective. Tiled Z stacks spanning all or most of the explant were captured every 15 min at 4 μm Z spacing.

## Modelling

All modelling was performed in R (*R Core Team, 2017*). Comparisons of the proportion of uninduced progenitors were based on the mean and standard deviation obtained from 10 simulations per condition. See Source Code File one for full implementation of the model. We developed a model to test the hypothesis that random migration with spatially defined commitment is sufficient to give rise to spatially restricted nephrogenesis. Cells are represented as single points in space that are positioned around a central 'tip centre' (0,0,0) in 3D space. As time progresses in 'steps' cells either make movements, or commit, based on a set of parameters described below:

**Random migration**: Each cells displacement has a random contribution that is dependent on a displacement value sampled from a distribution with mean 0, standard deviation equal to the *random migration value* and a random direction which is determined by the unit vector in the direction (x,y,z) where each component is sampled from the range −1, 1.

**Tip attraction**: Each cell is attracted toward the tip centre with a magnitude defined by a *tip attraction value* x (*tip_size^2 / (distance to tip)^2*), for *distance to tip >= tip size.*

**Tip repulsion**: At close range, where distance to tip <*tip size* cells are repelled from the tip with a magnitude of *tip attraction* value.

**Cell type**: Cells are defined as either 'cap', or 'committed'.

**Induction cutoff**: Cells within the induction zone, defined as all y < induction cutoff, have state count increased by one per step.

**State threshold**: Once cells reach the state threshold value they shift from 'cap' to' committed' type and cease moving.

**Cell division:** A proportion, defined by mitotic index, of cells of type 'cap' are randomly selected and duplicated per step (unless stated, the mitotic index value is 0.01)

**Cap re-entry**: Cells with some accumulated state count have this reduced by *re-entry* value for each step that they are outside the induction zone.

**Scaling**: Both tip size and induction cutoff are multiplied by the *niche scale factor*, allowing the entire niche architecture to be scaled down in size.

Simulation values for the 'primed' model were: *Tip size = 5, Induction cutoff = −2, Distance to tip = Euclidian 3D distance from the position of each cell to the tip centre, Tip attraction value = 0.5, Random migration value = 0.5, State threshold = 30, Mitotic index = 0.01, re-entry value = 0, Time steps = 400, Niche scale factor = 1.0.* Simulation values for the 'return' to ground model: *As above, except that re-entry value = 1.*

## Whole mount image analysis

Image quantification was performed using Fiji and R. Customs scripts employing the Laplacian of Gaussian based spot detector in the Trackmate library (*Tinevez et al., 2017*) of Fiji (with median filtering, sub pixel localisation and threshold of 3000) were used to detect the approximate position of each tdTomato[+] (radius = 5.0), or SIX2[+] (radius = 4.0) cell. This method resulted in robust detection of labelled cells, while excluding large labelled objects in the mature nephron. Signal intensity for each channel within the neighbourhood of each spot was outputted and imported into R for further analysis.

The proportion of tdTomato-labelled nephron progenitor cells was calculated by comparing tdTomato spot data with SIX2 spot data that had been filtered to remove cells with scaled value less than 0. Filtered datasets consisted thousands of spots per image (mean 15000 spots per image, minimum 6140). Nearest neighbour analysis in 3D space was used to identify the nearest neighbouring SIX2 spot for each tdTomato spot detection. tdTomato cells with a nearest neighbour SIX2 spot within 8 μm, and *Wnt4*-GFP signal less than 10000 were classed as being SIX2 +cap cells. This definition captured cells with high SIX2 intensity as well as including cap cells deeper in the sample with reduced SIX2 intensity due to imaging depth, but located close to a SIX2 spot detection. The proportion of labelled progenitors was defined as the number of SIX2 +labelled cap cells as a

proportion of the number of filtered SIX2$^+$ spot detections. For each time point between 3 and 5 wholemount kidneys from different embryos were quantified, except E14.5 labelling/E15.5 collection where two kidneys from different embryos were quantified.

For constitutive *Wnt4*-Cre labelling experiments YFP labelling was not sufficiently sparse to obtain satisfactory spot-based analysis. Instead we used the spot detection approach described above to quantify the entire population of SIX2$^+$ nuclei. These data were filtered to remove bright spurious detections (log SIX2 signal greater than 10). To define parameters for classification we visually marked the position of ~100–200 YFP labelled cap cells per dataset using Mamut (*Wolff et al., 2018*) and identified the same positions in automated spot count data based on nearest neighbour distance less than three microns. These manual reference points were used to calculate a nuclear (Draq5) cutoff value (minimum nuclear value in the manual dataset), SIX2 cutoff (minimum SIX2 value in the manual dataset) and YFP cutoff (Mean YFP value in the manual dataset, minus one standard deviation). Spots below the nuclear and SIX2 cutoff values were filtered, and remaining spots were classified as labelled vs unlabelled cap based on YFP cutoff value. For display SIX2 and GFP values were normalised to the 99$^{th}$ quantile value within each population. Analysis was performed on seven wholemount kidney datasets, from 7 embryos across two litters.

### Time lapse image analysis

Migration of *Wnt4*$^{GCE}$ (n = 6 datasets) or Six2$^{GCE}$ (n = 4 datasets) labelled cells in time lapse images were analysed using Fiji with the Trackmate plugin. We tracked all labelled cells that could be individually resolved, including cells that had recently undergone, or were currently undergoing re-entry into the nephron progenitor domain. A combination of automatic tracking and manual annotation was used to track cell and tip movement (381 tracks total). All tracks were visually inspected to ensure accuracy. Drift correction was performed using previously described methods (*Combes et al., 2016*; *Lefevre et al., 2016*). Further analysis of tracking data was performed in R. 3D projections for visualisation were generated using Imaris (Bitplane).

Analysis of cell fate in high resolution time lapse images (*Figures 3* and *6*) was performed using the Mamut plugin (*Wolff et al., 2018*) in Fiji and Imaris (BitPlane). Initially Mamut was used to manually track a landmark within the *HoxB7*-GFP ureteric epithelium, typically a tip, for each region of interest. Mamut was used to extract sub-volumes centered around the tracked *HoxB7*-GFP landmark, ensuring cell movement was viewed corrected for tip/sample movement and to provide a reference for concatenating time lapses where the microscope stage position had been adjusted. Sub-volumes were further analysed in Mamut (*Figure 3*), or Imaris (*Figure 6*) to manually track cell movement and generate visualisations. The final identity of tracked cells or regions was confirmed by examining IF staining of samples fixed immediately at the completion of live imaging. Time lapse end-points were matched to IF datasets manually using morphological landmarks. Given sparse tdTomato labelling in the cap mesenchyme of explant samples, unambiguous identification of tracked cells was trivial once the correct region had been identified.

### Generation of single-cell sequencing libraries

E15.5 embryonic kidneys were dissected and dissociated in 1 mg/ml liberase (Roche) on ice for 25 min. Samples were gently agitated by pipetting every 5 min. Samples were filtered, washed once with cold PBS and pelleted by centrifugation. Samples were resuspended at the appropriate concentration and stored on ice prior to the generation of single cell libraries using the 10x Chromium platform.

### Bioinformatics

The Cell Ranger pipeline (v1.3.1) was used to perform sample demultiplexing, barcode processing and single-cell gene counting (*Zheng et al., 2017*). Briefly samples were demultiplexed to produce a pair of FASTQ files for each sample. Reads containing sequence information were aligned to the mm10 reference genome with the additional of the R26 tdTomato and STOP transcripts. Cell barcodes were filtered to remove empty droplets and PCR duplicates were removed by selecting unique combinations of cell barcodes, UMIs and gene ids. The samples were aggregated to produce a gene expression matrix that was used for further analysis.

CellRanger output was imported into R using the Seurat library (v2.3.2) (*Butler et al., 2018*; *Satija et al., 2015*). Cell cycle prediction was performed using the Cyclone function within the scran R package (v1.6.7) (*Lun et al., 2016*; *Scialdone et al., 2015*). Quality control was performed in Seurat to remove cells with less than 200 genes expressed or with greater than 7.5% mitochondrial genes expression. Genes expressed in less than 3 cells were removed from the dataset. The filtered dataset contained 3451 cells (mean of 4166 genes detected per cell). Scaled data matrices were generated by regressing against the number of unique molecular identifies (UMIs), percentage of mitochondrial genes expressed and cell cycle prediction scores for G1, S and G2M phase. For whole kidney data SNN clustering was performed in Seurat using resolution 0.5 for with the first 15 principal components calculated from a set of 1201 variable genes. Differentially expressed marker genes of each cluster were identified in Seurat (Wilcoxon rank sum test) and used to assign clustering identify based on GO analysis (ToppFunn) and known canonical markers of each compartment. Markers that define change from progenitor to committed state were identified as the 30 most significantly differentially expressed between progenitor clusters (cluster 0, 1) and committed nephron clusters (cluster 2, 3, 5). tdTomato expressing cells were defined as those expressing greater than 0.5 log normalised counts using standard normalisation in Seurat.

## Acknowledgements

The authors thank Prof Andrew McMahon for access to the *Wnt4*^GCE line and Dr Hiroshi Kataoka, Prof Richard Harvey, Dr Aude Dorison for access to the *Pdgfra*CreER line. We acknowledge the assistance of Matthew Burton and the MCRI Imaging Facility and the services and facilities of AGRF. MHL is a Senior Principal Research Fellow of the National Health and Medical Research Council of Australia (GNT1136085). ANC was supported by a DECRA fellowship from the Australian Research Council (ARC) (DE150100652). LZ is supported by an Australian Government Research Training Program (RTP) Scholarship. This work was supported by the NHMRC (GNT1063989, GNT1156567), ARC (DE150100652), and funding to ANC from the Cell Biology theme, Murdoch Children's Research Institute.

## Additional information

### Competing interests

Melissa H Little: Has consulted for and received research funding from Organovo Inc. The other authors declare that no competing interests exist.

### Funding

| Funder | Grant reference number | Author |
|---|---|---|
| National Health and Medical Research Council | GNT1156567 | Alexander N Combes |
| Australian Research Council | DE150100652 | Alexander N Combes |
| Murdoch Children's Research Institute | | Alexander N Combes |
| National Health and Medical Research Council | GNT1136085 | Melissa H Little |
| National Health and Medical Research Council | GNT1063989 | Melissa H Little |

The funders had no role in study design, data collection and interpretation, or the decision to submit the work for publication.

### Author contributions

Kynan T Lawlor, Conceptualization, Software, Formal analysis, Validation, Investigation, Visualization, Methodology, Writing—original draft, Project administration, Writing—review and editing; Luke Zappia, Formal analysis, Writing—review and editing; James Lefevre, Formal analysis, Validation;

Joo-Seop Park, Resources, Formal analysis; Nicholas A Hamilton, Alicia Oshlack, Formal analysis, Supervision; Melissa H Little, Supervision, Funding acquisition, Writing—original draft, Project administration, Writing—review and editing; Alexander N Combes, Conceptualization, Supervision, Funding acquisition, Investigation, Methodology, Writing—original draft, Project administration, Writing—review and editing

### Author ORCIDs
Kynan T Lawlor  http://orcid.org/0000-0003-4080-5439
Luke Zappia  http://orcid.org/0000-0001-7744-8565
Nicholas A Hamilton  http://orcid.org/0000-0003-0331-3427
Alicia Oshlack  http://orcid.org/0000-0001-9788-5690
Alexander N Combes  http://orcid.org/0000-0001-6008-8786

### Ethics
Animal experimentation: All animal experiments were assessed and approved by the Murdoch Children's Research Institute Animal Ethics Committee (A783/A894) and were conducted in accordance with applicable Australian laws governing the care and use of animals for scientific purposes.

### Decision letter and Author response
Decision letter https://doi.org/10.7554/eLife.41156.029
Author response https://doi.org/10.7554/eLife.41156.030

## Additional files

### Supplementary files
• Source code 1. R code for computer simulation of stochastic commitment model.
DOI: https://doi.org/10.7554/eLife.41156.022

• Supplementary file 1. Marker gene lists for whole kidney single cell transcriptional data.
DOI: https://doi.org/10.7554/eLife.41156.023

• Supplementary file 2. Marker gene lists for nephron lineage single cell transcriptional data. scRNA-seq data is available via GEO accession GSE118486 (token kfwtoegufxejzkz).
DOI: https://doi.org/10.7554/eLife.41156.024

• Transparent reporting form
DOI: https://doi.org/10.7554/eLife.41156.025

### Data availability
Single cell sequencing data has been deposited in GEO under accession code GSE118486. Gene lists from the single cell analysis and code for the simulation of cell migration and stochastic commitment have been provided as Supplementary Files.

The following dataset was generated:

| Author(s) | Year | Dataset title | Dataset URL | Database and Identifier |
| --- | --- | --- | --- | --- |
| Lawlor KT, Zappia L, Lefevre J, Park J-S | 2019 | Single cell sequencing data from Nephron progenitor commitment is a stochastic process influenced by cell migration | https://www.ncbi.nlm.nih.gov/geo/query/acc.cgi?acc=GSE118486 | NCBI Gene Expression Omnibus, GSE118486 |

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
