## [Decision Letter]

Thank you for submitting your article "Nephron progenitor commitment is a stochastic process influenced by cell migration" for consideration by *eLife*. Your article has been reviewed by two peer reviewers, and the evaluation has been overseen by a Reviewing Editor and Marianne Bronner as the Senior Editor. The reviewers have opted to remain anonymous.

The reviewers have discussed the reviews with one another and the Reviewing Editor has drafted this decision to help you prepare a revised submission. Please aim to submit the revised version within two months.

Summary:

This work proposes an intriguing model to describe maintenance of the nephron progenitor cell niche during kidney development. Specifically, a combination of genetic lineage mapping and 3D cell movement analysis allow the authors to conclude that cells reaching the renal vesicle can return back to the cap mesenchyme. Furthermore, the data also point to an intermediate state in differentiation in which cells can commit to epithelial differentiation over a much longer period than previously thought. The study extends previous work describing the migration of cells within cap mesenchyme and adds the essential dimension of genetic lineage marking. This work supports a highly dynamic model for nephron progenitor differentiation than was previously appreciated, and it will provoke some reevaluation of our interpretation of cell signaling in this niche.

Essential revisions:

As you will see from the combined comments of the reviewers below, both agree that this is a valuable study, but have expressed concerns that the data in its current form does not definitively support the conclusion that *Wnt4* lineage-marked cells from an epithelial PTA to the cap contributes to the nephron progenitor pool (Figure 3 and Supplemental Videos 2 and 3) or that the cells from a single "aggregate (i.e., PTA) can contribute to multiple nephrons (Figure 6 and Supplemental Video 5). This is explicitly pointed out in the points raised below. We would be pleased to consider a revised version of your manuscript that includes better movies (and corresponding figures) that show cells actually starting in a PTA and ending up in cap mesenchyme (or multiple nephrons).

In revising your manuscript can you also attend to the large number of points raised by the reviewers, many of which are minor.

Major points Reviewer 1:

1) While the data in Figures 1 and 2 convincingly show that tdTomato+, *Wnt4* lineage-labeled cells, presumably labeled in the PTA, are later found in the SIX2+ cap mesenchyme (nephron progenitor population), the time lapse imaging in Figure 3 and Supplemental Videos 2 and 3 does not support this conclusion. In Video 2, none of the cells are marked and it is not clear which cells the reader is meant to follow. In Video 3, and the corresponding Figure 3C and D, several cells are tracked, but it does not seem to me that they start in a PTA, as the authors claim. They start near a UB tip, but what is the evidence that they are in a PTA? If these are the best examples they can find, it casts some doubt on the conclusion.

2) The difference in mean migration speed of Six2Cre-labeled cells and *Wnt4*-Cre-labeled cells may be statistically significant, but the difference of the mean speeds is so small (and the variance so large) that I doubt its biological significance. Also, the data are corrected "for the movement of the adjacent GFP-marked epithelial tip" – since the tip is a structure that is growing and changing shape, I'm not sure that this is a valid correction.

3) I find the model in Figure 4 to be very confusing (particularly for the general scientific audience of *eLife*). It adds little to the paper and the authors should consider greatly simplifying or removing it.

4) Figure 6 and Supplemental Video 5 are intended to show that *Wnt4*-lineage labeled "cells from these initial aggregates contributed to several adjacent nephrons, counter to the established understanding that a single induction event gives rise to a single nephron". This would be a surprising result, but it is not supported by the data. When I examined the starting Tomato+ cells that are tracked in Figure 6B-F and Video 5, the tracks do not start out in a single aggregate but in a broad region surrounding one of the ureteric bud tips. The same is true of the experiment shown in Figure 6A – the starting cells at 0 hrs (shown in the inset) do not appear to be only in PTAs, but are widely dispersed around the ureteric tips. It is likely that some of these cells are NP cells in the cap mesenchyme at the start of the movie, and it is not surprising that they later contribute to multiple nephrons. The authors write "Groups of *Wnt4* lineage cells that were immediate neighbours at the onset of imaging ultimately contributed to multiple adjacent nephrons". However, the cells shown in the figures were not "immediate neighbours" but were widely dispersed around one or more tips.

Major Points: Reviewer 2:

1) Overall, the description of the localization of *Wnt4*-GFP-creERT localization is insufficient. For example, ISH shows strong expression of *Wnt4* in stroma and the authors comment on discrepancies with *Wnt4-*Cre in the text. Without having a better understanding of whether or not expression is limited to the PTA/RV, it is not possible to be definitive about where the labeled cells that enter the cap are coming from. In the RNA-Seq section, the authors mention that they see cells co-expressing *Cited1* and *Wnt4* and interpret this as cells that are in transition between states. This is an assumption that rests on the immunostaining data, and highlights a potential confounder in the experiment: Could the tdTomato labeling in the cap mesenchyme be partly or entirely due to transitory *Wnt4*-cre expression in cap mesenchyme cells that is insufficient to cause them to fluoresce? In the quantitative analysis there are quite a number of Tomato+ SIX2+ cells only 24 hours after tamoxifen pulse (Figure 2A). This point needs to be addressed and it really needs to be discussed at the very beginning of the manuscript because it is foundational for the whole interpretation.

2) The E10.5 pulse chase data are difficult to understand because it appears incomplete. In which cells is *Wnt4*-cre expressed at E10.5? Marker co-staining would be required to define the cell types(s) in which expression is seen. 2H-J: it is not possible from the data presented to conclude that labeling is seen in PTA and RV; the red cells do not look epithelial and they mostly appear to localize outside the GFP+ clusters in 2I. Related to this, the inclusion of a tamoxifen-treatment at E10.5 for the RNA-seq experiment is problematic in my opinion because we do not know the identity of cells labeled at this stage. Could the authors explain the motivation behind labeling at a stage when there is no PTA/RV for a study of the characteristics of cells that recirculate from the PTA/RV?

3) The *Wnt4*-cre data gives an idea of the magnitude of recirculation of RV/PTA cells back to the cap mesenchyme. But it is hard to interpret because the staining cannot distinguish between active *Wnt4* expression (GFP) and historical *Wnt4* expression (YFP). Could the authors please show the staining in *Wnt4-*Cre kidneys without the YFP reporter?

4) Regarding the time-lapse data, the authors mention in the text that the GFP signal from the *Wnt4*-cre is too weak to detect, and the GFP seen in the pictures is only from the *Hoxb7*-GFP. It is therefore very difficult to orient the 3D pictures, especially the movies which are very helpful to understand cell migration. Could the authors provide some landmarks for orientation?

5) During the past few years many investigators have shown strong effects of tamoxifen on various aspects of cell biology and it is not out of the question that it could be influencing migratory behavior in the cap mesenchyme. It would be valuable to control for this by comparing the effect of tamoxifen treatment versus vehicle (especially the repeated doses used in some experiments) on recirculation of *Wnt4-*Cre-labeled cells back into cap mesenchyme.

6) The language used to describe magnitudes is very vague in places, so it is difficult for the reader to interpret the importance of the findings. For example "several" Tomato+ cells express *Six2* but not *Wnt4*. This is not very helpful and a percentage would be appropriate here.

7) For the E11.5 explant labeling in Figure 6, it is quite difficult from the text to understand exactly what the authors have done. Also, the referenced Supplemental Video 6 is missing. There is no marker for cap mesenchyme and the explant morphology is not similar to in vivo kidney, so the statement that Tomato+ cells contribute to cap mesenchyme is not supported. The central data here is the assignment of cells to a particular PTA/RV and then the tracing of those cells back out to multiple RVs, but those data are not presented so it is not possible to evaluate this experiment.

[Editors' note: further revisions were requested prior to acceptance, as described below.]

Thank you for resubmitting your work entitled "Nephron progenitor commitment is a stochastic process influenced by cell migration" for further consideration at *eLife*. Your revised article has been favorably evaluated by Marianne Bronner as the Senior Editor, and a Reviewing Editor.

The manuscript has been improved but there is one remaining issue that needs to be addressed before acceptance, as outlined below:

The issue concerns the definition of a PTA, and whether cells actually leave a PTA to migrate and revert to the NP state. The traditional view of a PTA is a cluster of cells in the process of aggregating to form an RV – not yet epithelial, but more closely aggregated than NP cells and committed to forming the RV. In your manuscript you clearly suggest that some cells leave the PTA. The reviewers propose an alternative model that you might like to consider. They note that most of the *Wnt4*+ cells under the UB tip aggregate to form the PTA, and this is an irreversible step, but a few closely neighboring *Wnt4*+ cells don't incorporate into the aggregate allowing them to migrate away and revert to the NP state. The *Wnt4*-GFP positive cells in Figure 1G-K seem to be mostly aggregated, but there seem to be a few *Wnt4*+ cells around the periphery of the PTA that may not be part of the aggregate. Accordingly the data don't support the interpretation that cells clearly in an aggregate are the ones that revert to NP. While we note this is a subtle point, it is important for the definition of a pretubular aggregate. Hopefully you can tweak the final version of the text accordingly.

---

## [Author Response]

Essential revisions:As you will see from the combined comments of the reviewers below, both agree that this is a valuable study, but have expressed concerns that the data in its current form does not definitively support the conclusion that Wnt4 lineage-marked cells from an epithelial PTA to the cap contributes to the nephron progenitor pool (Figure 3 and Supplemental Videos 2 and 3) or that the cells from a single "aggregate (i.e., PTA) can contribute to multiple nephrons (Figure 6 and Supplemental Video 5). This is explicitly pointed out in the points raised below. We would be pleased to consider a revised version of your manuscript that includes better movies (and corresponding figures) that show cells actually starting in a PTA and ending up in cap mesenchyme (or multiple nephrons).In revising your manuscript can you also attend to the large number of points raised by the reviewers, many of which are minor.

We thank the reviewers for these comments. We would start, however, by noting that the pretubular aggregate is not an epithelial structure and that this assumption significantly impacts on the interpretation of our data and proposed model. To make this point and to respond to these major concerns, we have generated new data and cite additional sources to clarify that a pretubular aggregate (PTA) is a cluster of cells, located at the tip-stalk junction of the ureteric epithelium that express key markers including *Wnt4*, and lack definitive epithelial properties such as polarised distribution of aPKC (Figure 1). We hope that this clearly makes the point that we are not claiming that cells can escape from an epithelial structure such as a renal vesicle. We have also added more data to show that labelling of *Wnt4*-expressing cells only occurs in the PTA and not in the cap mesenchyme but that escaping cells can repopulate the cap mesenchyme niche over time (Figures 1 and 2). New, higher resolution, live and fixed data has been generated to support the movement of cells from the region where *Wnt4* is expressed and nephrons form, back into the nephron progenitor pool (Figure 3). We have also clarified the eventual contribution of primed cells to multiple nephrons in the early developing kidney (Figure 6) as detailed below. Principally, these data demonstrate that nephron progenitor commitment is not unidirectional and unveils a greater plasticity in this dynamic population than was previously appreciated.

Major points Reviewer 1:1) While the data in Figures 1 and 2 convincingly show that tdTomato+, Wnt4 lineage-labeled cells, presumably labeled in the PTA, are later found in the SIX2+ cap mesenchyme (nephron progenitor population), the time lapse imaging in Figure 3 and Supplemental Videos 2 and 3 does not support this conclusion. In Video 2, none of the cells are marked and it is not clear which cells the reader is meant to follow. In Video 3, and the corresponding Figure 3C and D, several cells are tracked, but it does not seem to me that they start in a PTA, as the authors claim. They start near a UB tip, but what is the evidence that they are in a PTA? If these are the best examples they can find, it casts some doubt on the conclusion.

We present a substantially revised dataset including new videos, more support and more careful description of where cells start and end.

We interpret a PTA as a group of cells located at the tip-stalk junction, that express *Wnt4* and other key markers. We draw on detailed studies of early nephron epithelialisation, classic and recent papers describing PTAs to support this interpretation. As such, a cell that is located under the tip and expresses *Wnt4* is arguably in a PTA, or a further stage of commitment towards an epithelial nephron.

As acknowledged, we have presented fixed imaging in Figures 1-3 and supplements to support that nephron lineage *Wnt4*-expressing cells are labelled in the PTA/tip-stalk junction, not in the nephron progenitor population, and that these cells appear in the cap mesenchyme over time. These in vivo lineage tracing data are then supported by our live imaging, which observes part of that journey in real time.

We present a revised Figure 3 and accompanying text that illustrates analogous initial labelling of early committing nephron structures in vivo (Figure 3C-E) and in our culture system (Figure 3F). In live imaging of similar samples over 24 hours we show *Wnt4*-lineage cells moving from the region under the tip to the top or end of a tip (Figure 3G, J). Fixed analysis of these samples after live imaging shows that the tdTomato-labelled cells have migrated back into the cap mesenchyme and are surrounded by unlabelled SIX2-expressing cells (Figure 3H, K). The region under the tip that these cells migrated from contains a population of cells that have more prominent basolateral localisation of NCAM and lower levels of SIX2, suggesting those cells are undergoing the early stages of epithelialisation to form a nephron, similar to what we would expect in a PTA progressing to a primitive renal vesicle.

2) The difference in mean migration speed of Six2Cre-labeled cells and Wnt4-Cre-labeled cells may be statistically significant, but the difference of the mean speeds is so small (and the variance so large) that I doubt its biological significance. Also, the data are corrected "for the movement of the adjacent GFP-marked epithelial tip" – since the tip is a structure that is growing and changing shape, I'm not sure that this is a valid correction.

We agree that the difference is small and have revised the corresponding text to reflect this. These figure panels now form a figure supplement for Figure 3.

Regarding the second point, assessing cell movement relative to the ureteric tip is necessary to distinguish between displacement caused by cell migration and that caused by bulk tissue movement. This is a common practice in tracking cell movement in developing tissues and has been published previously for the developing kidney (Combes et al., 2016). For example, consider the movement of a nephron progenitor cell that is attached to the ureteric tip through cell-cell adhesion. When the ureteric tip grows and branches the nephron progenitor cell will be moving simply because it is attached to the tip epithelium. This gives the impression of cell movement when, relative to the tip, the nephron progenitor cell has not moved at all.

3) I find the model in Figure 4 to be very confusing (particularly for the general scientific audience of eLife). It adds little to the paper and the authors should consider greatly simplifying or removing it.

We have clarified the text describing this model and the conclusions arising from it. We believe this is a valuable addition to the manuscript and that modelling complements and extends experimental studies. *eLife* frequently publishes articles that integrate computational biology and development.

4) Figure 6 and Supplemental Video 5 are intended to show that Wnt4-lineage labeled "cells from these initial aggregates contributed to several adjacent nephrons, counter to the established understanding that a single induction event gives rise to a single nephron". This would be a surprising result, but it is not supported by the data. When I examined the starting Tomato+ cells that are tracked in Figure 6B-F and Video 5, the tracks do not start out in a single aggregate but in a broad region surrounding one of the ureteric bud tips. The same is true of the experiment shown in Figure 6A – the starting cells at 0 hrs (shown in the inset) do not appear to be only in PTAs, but are widely dispersed around the ureteric tips. It is likely that some of these cells are NP cells in the cap mesenchyme at the start of the movie, and it is not surprising that they later contribute to multiple nephrons. The authors write "Groups of Wnt4 lineage cells that were immediate neighbours at the onset of imaging ultimately contributed to multiple adjacent nephrons". However, the cells shown in the figures were not "immediate neighbours" but were widely dispersed around one or more tips.

We have revised the presentation of these data to focus on a group of six cells that are initially 1-3 cell diameters apart and located under the ureteric tip in a field of *Wnt4*-labelled cells. The spacing between these cells could reflect partial labelling of a broader domain of *Wnt4*-expressing cells; we rarely see expected domains of *Wnt4* expression being completely labelled. We provide unambiguous tracking of these six cells with the context of a rendered tip surface to show that they remain under the tip, near the region that an early nephron would form. Instead of forming a renal vesicle in place, these six *Wnt4*-lineage cells end up contributing to three distinct nephrons. As such, these data do support that a single induction event can lead to a partially committed population that can seed multiple nephrons. As stated by the reviewer this is a surprising result as we would expect all labelled cells to immediately progress to form a nephron. Any new nephrons would presumably arise from new commitment events that would occur after the tamoxifen has washed out, and therefore should not be labelled. However, we now qualify these results, acknowledging that they arise from E11.5 kidneys, which have large domains of *Wn4* expression and large tips. The dynamics of nephron induction and commitment may be different at later stages of development, which will be tested in future studies.

Major Points: Reviewer 2:1) Overall, the description of the localization of Wnt4-GFP-creERT localization is insufficient. For example, ISH shows strong expression of Wnt4 in stroma and the authors comment on discrepancies with Wnt4-Cre in the text. Without having a better understanding of whether or not expression is limited to the PTA/RV, it is not possible to be definitive about where the labeled cells that enter the cap are coming from. In the RNA-Seq section, the authors mention that they see cells co-expressing Cited1 and Wnt4 and interpret this as cells that are in transition between states. This is an assumption that rests on the immunostaining data, and highlights a potential confounder in the experiment: Could the tdTomato labeling in the cap mesenchyme be partly or entirely due to transitory Wnt4-cre expression in cap mesenchyme cells that is insufficient to cause them to fluoresce? In the quantitative analysis there are quite a number of Tomato+ SIX2+ cells only 24 hours after tamoxifen pulse (Figure 2A). This point needs to be addressed and it really needs to be discussed at the very beginning of the manuscript because it is foundational for the whole interpretation.

The localisation of the *Wnt4*GFPCreERT2 is consistent with the established sites of expression of *Wnt4* – i.e. highest in the early committing nephron (PTA/RV), and expressed at lower levels in some stromal cells, particularly in the medulla (Georgas et al., 2009). We have added new data to Figure1—figure supplement 1 to illustrate the sites of expression in the inducible *Wnt4*^GCE^ line. The cell labelling driven by this Cre preferences labelling in pretubular aggregates and renal vesicles, and only labels stromal cells with higher doses of tamoxifen. To address whether stromal *Wnt4*-lineage cells contribute to the nephron progenitor population we performed lineage tracing with an inducible *Pdgfra* cre. *Pdgfra* is expressed in all stromal cells in the developing kidney, including those that express *Wnt4* (DiRocco et al., 2013). Lineage tracing with this inducible Cre results in extensive stromal labelling but does not result in labelled cells within the cap mesenchyme. As such, we are sure that these *Wnt4*-labelled cells arise in the PTA, not from the stroma. We now include this additional lineage tracing data in Figure 1—figure supplement 1.

With regard to the possibility of a transient pulse of *Wnt4* expression in nephron progenitor cells- we considered this and carefully looked for such events. We comprehensively analysed the location of tdTomato labelled cells 24 hours after addition of tamoxifen at E12.5 or E14.5. As reported, labelled cells were always observed within clusters of *Wnt4*-GFP expressing cells at the tip-stalk junction. No labelled nephron progenitor cells were observed in the upper region of the cap at 24 hours after induction suggesting that these cells are not being labelled directly. We have example images to support this conclusion in Figures 1, 2, 3 and Figure 2—figure supplement 1.

The data in revised Figure 2C shows that ~1% of the tdTomato labelled cells express low levels of SIX2. This is generated from quantitative analysis of wholemount immunofluorescence data. When we visualise these SIX2^+^GFP^+^ cells in the 3D data, they are located in in the early committing nephron. This is consistent with the immunofluorescence we show in Figure 1G and Figure 2B, where labelled SIX2^+^GFP^+^ cells can be seen at the tip-stalk junction in early committing nephron structures.

2) The E10.5 pulse chase data are difficult to understand because it appears incomplete. In which cells is Wnt4-cre expressed at E10.5? Marker co-staining would be required to define the cell types(s) in which expression is seen. 2H-J: it is not possible from the data presented to conclude that labeling is seen in PTA and RV; the red cells do not look epithelial and they mostly appear to localize outside the GFP+ clusters in 2I. Related to this, the inclusion of a tamoxifen-treatment at E10.5 for the RNA-seq experiment is problematic in my opinion because we do not know the identity of cells labeled at this stage. Could the authors explain the motivation behind labeling at a stage when there is no PTA/RV for a study of the characteristics of cells that recirculate from the PTA/RV?

In our revision we have removed focus from the E10.5 labelling as it is not central to the main messages of the paper. These data are now presented as a figure supplement.

To address the questions: SIX2+ cells are specified in the intermediate mesoderm around E8.5 (Taguchi et al., Cell Stem Cell 2014) and there is no published data to suggest the contribution of additional cell types to the *Six2* lineage after this stage, although this remains a possibility. Lineage tracing from an inducible *Six2*Cre suggests the nephron progenitor population self-renews from E10.5 onwards with no additional input from any other cell population (Kobayashi et al.,2008). Within this context, our results suggest *Wnt4* is expressed in nephron progenitor cells within the metanephric mesenchyme at E10.5 because cells labelled from induction at that time co-express nephron progenitor marker SIX2 and *Wnt4*-GFP at E11.5 (Figure 2—figure supplement 3).

Regarding the localisation of cells in the previous version of Figure 2H-J; we agree that several labelled cells are not contained within RV or PTA structures in this image, however many cells are contained within clusters of cells expressing *Wnt4*-GFP under the ureteric tip that resemble PTA and epithelializing structures.

Including E10.5 labelling in the scRNA-Seq experiment aimed to maximize the number of cells labelled for analysis. As explained above, the E10.5 induction primarily labelled *Wnt4*-GFP+ SIX2+ cells at E11.5, allowing us to interrogate the early stages of nephron formation. This study has focussed on how nephron progenitor cells commit to form a nephron. We show that not all cells that are induced to express *Wnt4* undergo epithelialisation or integrate into a nephron. This early stage is part of an experiment to determine whether cells that have previously expressed *Wnt4* have any lasting transcriptional changes associated with that event. We delivered tamoxifen at 10.5, 11.5, 12.5 and 13.5 to label cells across this developmental range. The cells arising from the 10.5 labelling make up a fraction of the final population that was analysed but still serve to support the conclusion that nephron progenitor cells can return to an uncommitted state after being induced to express *Wnt4*.

3) The Wnt4-cre data gives an idea of the magnitude of recirculation of RV/PTA cells back to the cap mesenchyme. But it is hard to interpret because the staining cannot distinguish between active Wnt4 expression (GFP) and historical Wnt4 expression (YFP). Could the authors please show the staining in Wnt4-Cre kidneys without the YFP reporter?

The expected expression pattern in all mouse lines is illustrated by in situhybridisation results shown in Figure 1—figure supplement 1 and prior publications (Georgas et al., 2009).

4) Regarding the time-lapse data, the authors mention in the text that the GFP signal from the Wnt4-cre is too weak to detect, and the GFP seen in the pictures is only from the Hoxb7-GFP. It is therefore very difficult to orient the 3D pictures, especially the movies which are very helpful to understand cell migration. Could the authors provide some landmarks for orientation?

We have included additional images, labels and descriptions to clarify this.

5) During the past few years many investigators have shown strong effects of tamoxifen on various aspects of cell biology and it is not out of the question that it could be influencing migratory behavior in the cap mesenchyme. It would be valuable to control for this by comparing the effect of tamoxifen treatment versus vehicle (especially the repeated doses used in some experiments) on recirculation of Wnt4-Cre-labeled cells back into cap mesenchyme.

We are unable to control for the effect of tamoxifen in our experiments with the inducible Cre as we rely on this compound to induce labelling of the cells to monitor. However, we observe cells reintegrating into the nephron progenitor domain using a constitutive *Wnt4*-Cre (Figure 2), which does not rely on tamoxifen. The number of cells that reintegrate into the niche in the constitutive *Wnt4*-Cre is higher than that of the tamoxifen labelled samples, presumably because it is continuously active.

6) The language used to describe magnitudes is very vague in places, so it is difficult for the reader to interpret the importance of the findings. For example "several" Tomato+ cells express Six2 but not Wnt4. This is not very helpful and a percentage would be appropriate here.

We have included a quantitative measure as requested.

7) For the E11.5 explant labeling in Figure 6, it is quite difficult from the text to understand exactly what the authors have done. Also, the referenced Supplemental Video 6 is missing. There is no marker for cap mesenchyme and the explant morphology is not similar to in vivo kidney, so the statement that Tomato+ cells contribute to cap mesenchyme is not supported. The central data here is the assignment of cells to a particular PTA/RV and then the tracing of those cells back out to multiple RVs, but those data are not presented so it is not possible to evaluate this experiment.

As per response 4, we have revised the presentation of these data in Figure 6.

[Editors' note: further revisions were requested prior to acceptance, as described below.]

The manuscript has been improved but there is one remaining issue that needs to be addressed before acceptance, as outlined below:The issue concerns the definition of a PTA, and whether cells actually leave a PTA to migrate and revert to the NP state. The traditional view of a PTA is a cluster of cells in the process of aggregating to form an RV – not yet epithelial, but more closely aggregated than NP cells and committed to forming the RV. In your manuscript you clearly suggest that some cells leave the PTA. The reviewers propose an alternative model that you might like to consider. They note that most of the Wnt4+ cells under the UB tip aggregate to form the PTA, and this is an irreversible step, but a few closely neighboring Wnt4+ cells don't incorporate into the aggregate allowing them to migrate away and revert to the NP state. The Wnt4-GFP positive cells in Figure 1G-K seem to be mostly aggregated, but there seem to be a few Wnt4+ cells around the periphery of the PTA that may not be part of the aggregate. Accordingly the data don't support the interpretation that cells clearly in an aggregate are the ones that revert to NP. While we note this is a subtle point, it is important for the definition of a pretubular aggregate. Hopefully you can tweak the final version of the text accordingly.

Thank you for the feedback and opportunity to revise our work. We have included an additional paragraph in the Discussion incorporating the possibility that cells which return to the cap have been excluded from forming an aggregate. Our team had discussed this possibility previously but did not include it in the manuscript. The new paragraph is copied below.

"Our data support that Wnt4 expression immediately precedes the formation of a committed aggregate. […] The events that define the point at which cap re-entry is no longer possible are not well understood and may involve a gradual increase in adhesion, or a sharp switch from motile individual cells to an aggregated state."

We have carefully checked the rest of the paper and believe that there are no statements that explicitly say that cells from the PTA return to the NP population. As such we believe that the additional paragraph should satisfy the reviewer’s feedback.